statistics/pattern recognition/artificial intelligence

rape reporting delays, machine learning, spatial analysis, sexual violence, urban informatics

**Author for correspondence:**
Konstantin Klemmer
e-mail: k.klemmer@warwick.ac.uk

# Understanding spatial patterns in rape reporting delays

Konstantin Klemmer[1,2,3], Daniel B. Neill[3,4,5] and Stephen A. Jarvis[2,6]

[1]Department of Computer Science, University of Warwick, Coventry, UK
[2]The Alan Turing Institute, London, UK
[3]Center for Urban Science and Progress, New York University, New York, USA
[4]Courant Institute of Mathematical Sciences, New York University, New York, USA
[5]Robert F. Wagner Graduate School of Public Service, New York University, New York, USA
[6]College of Engineering and Physical Sciences, University of Birmingham, Birmingham, UK

(iD) KK, 0000-0002-7096-0133

Under-reporting and delayed reporting of rape crime are severe issues that can complicate the prosecution of perpetrators and prevent rape survivors from receiving needed support. Building on a massive database of publicly available criminal reports from two US cities, we develop a machine learning framework to predict delayed reporting of rape to help tackle this issue. Motivated by large and unexplained spatial variation in reporting delays, we build predictive models to analyse spatial, temporal and socio-economic factors that might explain this variation. Our findings suggest that we can explain a substantial proportion of the variation in rape reporting delays using only openly available data. The insights from this study can be used to motivate targeted, data-driven policies to assist vulnerable communities. For example, we find that younger rape survivors and crimes committed during holiday seasons exhibit longer delays. Our insights can thus help organizations focused on supporting survivors of sexual violence to provide their services at the right place and time. Due to the non-confidential nature of the data used in our models, even community organizations lacking access to sensitive police data can use these findings to optimize their operations.

## 1. Introduction

Recent years have seen a sharp rise in public awareness for sexual abuse issues. Under the #MeToo hashtag, people around the globe have shared their experiences of sexual violence. The reach of this campaign, extended through social media outlets such as Twitter and Facebook, underscores the grave burdens of sexual violence and its stigmatization around the world [1]. According to the

National Sexual Violence Resource Center (NSVRC), one in five American women will experience rape at some point in their lives [2]. This not only leads to unimaginable personal tragedy and trauma, but also carries immense economic costs for survivors,[1] with the lifetime cost of experiencing rape estimated at over US$120 000 [6]. In 1996, the overall annual cost of rape crime in the USA was estimated to be US$127 billion, more than that of any other crime [7]. As for per-crime costs, constituting the aggregate of costs for the survivor and society, several studies have estimated the cost of a rape crime to range between US$150 000 and US$283 000 [8–10]. Another significant problem is that survivors often do not report sexual assault, particularly rape, to the police: a recent survey by the US Department of Justice estimates that only one in every four rapes is reported [11]. Survivors cite various reasons they have chosen not to disclose sexual violence, ranging from fear of retaliation, to feelings of shame, post-traumatic stress, re-traumatization and mistrust in authorities. Further, many survivors who choose to report rape crimes only do so after a substantial delay after the crime occurrence. It is this delayed reporting which we seek to examine further in this study.

There are multiple reasons for investigating reporting delays of rape. First, unlike data gathered for under-reporting, data for delayed reporting of rape is available in relatively large quantities, via publicly available crime reports. Additionally, previous research has shown that longer reporting delays are a close proxy for under-reporting of rape [12]. This implies that by identifying subgroups vulnerable to delayed reporting, one would simultaneously also identify subgroups prone to under-reporting. Second, long reporting delays between crime occurrence and reporting also have significant implications: survivors who delay reporting are often left without professional support and care in the traumatic days directly after the rape, which can have severe mental and physical health consequences [13]. Moreover, the longer it takes for rape to be reported, the more complicated police investigations and perpetrator prosecution become, as key evidence, e.g. 'rape kits' become inapplicable. Furthermore, police officers have been shown to exhibit bias and distrust against rape survivors and particularly against delayed reporters, leading to the stigmatization of sexual violence survivors [14]. In 2018, the CNN investigation 'Destroyed' revealed that police across the USA had mishandled and destroyed 'rape kits' long before the expiration of the statutes of limitations, often for the most marginalized survivors [15]. Recent years have seen efforts by legislators and police to acknowledge and tackle such biases and to regain the trust of survivors, particularly those from disadvantaged socio-economic or ethnic backgrounds [16]. For example, promising approaches include specially trained police units [17] and close collaboration between police and external public-health organizations in survivor support [18].

Previous studies have identified predictors of delayed reporting and non-reporting; however, these findings are relatively inconsistent. The most important predictors in these studies have been crime- or survivor-specific, such as: infliction or threat of violence, relation to the perpetrator, survivor age and survivor education [19,20]. Very few studies have investigated area-level predictors, examining the impact of characteristics such as urban versus rural areas, Western versus non-Western countries or crimes committed indoors versus outdoors [21,22]. However, to our knowledge, no study has ever explicitly modelled the spatial context of rape reporting delays. Similarly, the effect of temporal characteristics on reporting behaviour has, to our knowledge, only been addressed by one study, which examined crimes committed during daytime versus night-time [22]. Spatial and temporal dependencies of reporting delays thus remain largely unexplored, making this study the first of its kind. Our paper also introduces several more demographic predictors which have never been used to study rape reporting delays, such as household structure and composition or languages spoken at home.

We show that rape reporting delays exhibit substantial spatial correlation, both at the event and area levels. These observations represent a novel finding on its own and are also the key motivation for the design of our methodological framework. By examining the residual spatial autocorrelation (RSA) from multiple spatial and non-spatial predictive models, we find that adjusting for survivor characteristics and temporal features can explain a substantial fraction of the variation in rape reporting delays, but does not fully eliminate spatial correlation. However, additionally controlling for the socio-economic and demographic features of the local area, or explicitly modelling spatial effects using the spatial coordinates of each case, both removes the RSA and improves predictive accuracy. Our research also introduces previously unutilized predictor variables, including new temporal features related to holidays. Most critically, in contrast to previous research, we focus on reporting delays—as opposed to

---

[1]Please note that throughout the text we refer to persons who reported rape crimes committed against their person as 'survivors' rather than 'victims'. Based on the literature, we believe this term better reflects the preferred self-identification of rape survivors. We would also like to point to the ongoing scientific discussion on the differences between the terms [3–5].

non-reporting—allowing us to analyse large, public datasets with fine-grained spatial information, which were not previously used in this domain.

The present study aims to build a predictive modelling framework for rape reporting delays. Over the last years, machine learning methods have become increasingly popular in crime research [23–25], particularly focusing on the spatial, temporal or spatio-temporal dimensions. For instance, recent studies have approached issues such as forecasting domestic violence [26], spatio-temporal contagion of gun violence [27] and prediction of criminal activity [28]. The rise in popularity of these methods can be explained in part by the increasing availability of large, often public, crime data and the simultaneous development of powerful, highly scalable machine learning algorithms. We seek to leverage the same advances to focus on the previously neglected issue of rape reporting delays, using publicly available data from the US cities of New York and Los Angeles. Taking these two cities as case studies, we highlight how combining emerging open data sources with scalable machine learning methods can help examine disparities and inform policy making to address them.

The contributions of our research are as follows:

1. This is the first large-scale study to investigate rape reporting delays using publicly available crime reports. In particular, our study is the first to focus on the spatial distribution of reporting delays, providing evidence for substantial spatial variation.
2. This research is also the first to apply predictive modelling to rape reporting delays. We examine the opportunities and challenges of applying predictive modelling methods to this specific problem.
3. Our findings shed new light on the process of rape reporting. Using only publicly available data and without access to detailed police reports, we build predictive models that explain a substantial amount of the variance in rape reporting delays. Additionally, we derive several predictor variables which have never been used before and assess their importance.
4. We establish a comprehensive analysis framework which may enable targeted, data-driven decision making and policy design. We identify features associated with delayed reporting and provide characteristics of areas which exhibit higher reporting delays. This way, we can identify vulnerable subpopulations which may benefit most from interventions such as educational campaigns or free counselling services. Our findings can help organizations to provide their services at the right place and time. Due to the open nature of our data, our approach can benefit both public authorities and non-governmental support organizations.
5. We highlight the importance of open data for crime research. Within the broader scope of machine learning for social good, our research shows how quantitative modelling may help our understanding of social processes and motivate evidence-based policies that aim at improving the well-being of individuals and communities.

The first section of this paper describes our methodology, data processing and machine learning models in detail, highlighting model choices, assumptions and technical design. The second section presents our results, beginning with a descriptive data analysis followed by the main findings from our modelling framework. We then discuss the implications of our findings, commenting on related literature and policy implications. We also address the limitations of our current approach and plans for future work.

# 2. Material and methods

## 2.1. Data

This study was conducted using public crime reports, accessed for the New York Police Department (NYPD) and Los Angeles Police Department (LAPD) jurisdictions using the cities' respective open data portals. A crime subset was then created based on the following criteria:

(a) It includes only rape crime, as classified by the respective police department, and excludes attempted rape.
(b) It includes only rapes committed in a domestic setting, as classified by the place of occurrence (residential building).
(c) It includes only rapes reported between 2013 and 2018.

This strengthens the assumption that local socio-demographic characteristics are representative of the survivors and perpetrators involved in the crimes. However, it is important to acknowledge that these

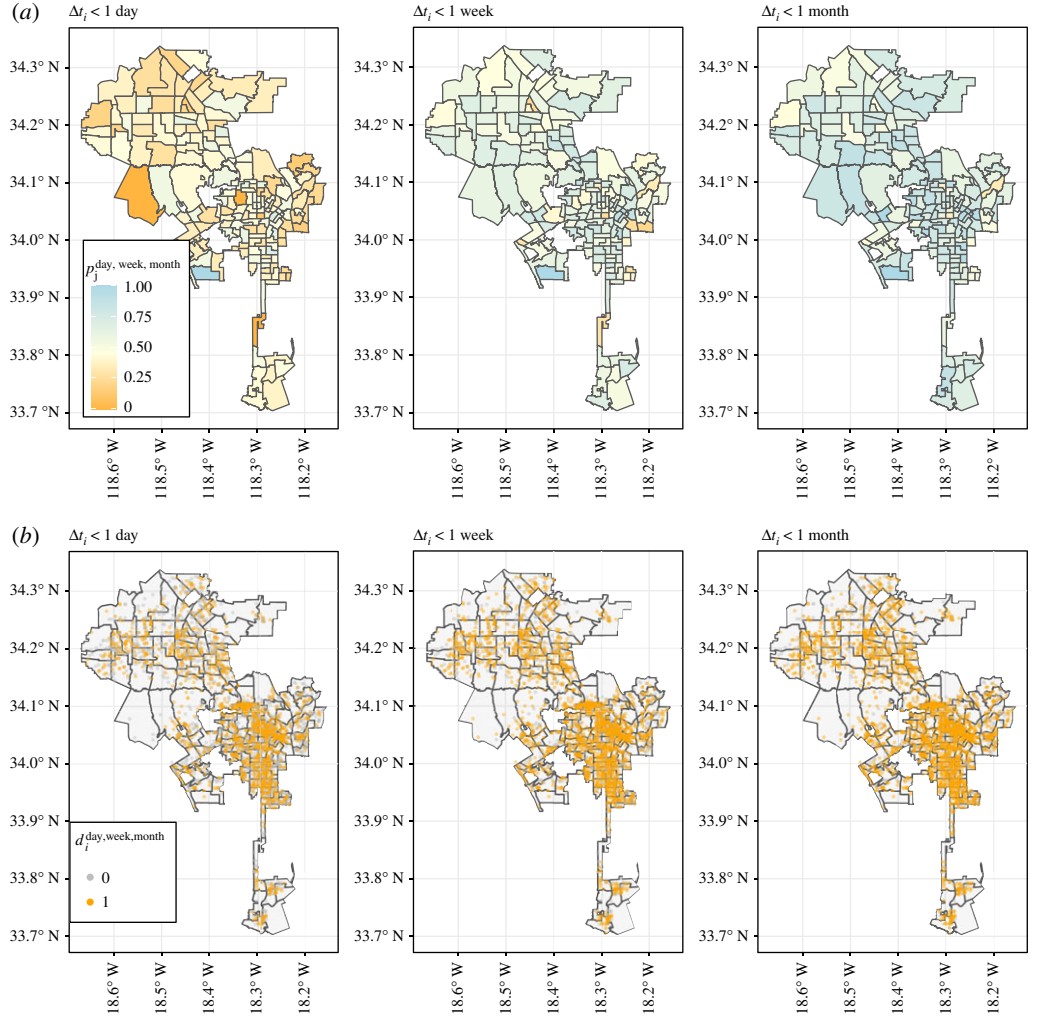

**Figure 1.** Spatial distribution of rape reporting delays in Los Angeles. (*a*) Share of rapes reported within one day, one week or one month across LAPD districts. (*b*) Highlighted in orange are rapes not reported within one day, one week or one month, respectively.

characteristics might be biased towards rapes that are eventually reported and available in the data, as opposed to all rapes, including the ones that are never reported or are reported outside the time frame of the available data. Nevertheless, the prior literature suggests that under-reporting and delayed reporting are highly correlated [9], and thus we would not expect the inclusion of these unreported rape crimes to substantially change the associations between predictors and target variables. We select data for the past 5 years only, assuming that over this period, socio-economic characteristics of the observed areas did not change substantially.

This results in a set of $n$ rape crimes, where each crime is reported in one of $m$ police districts, for each of the cities NY ($n = 5928$, $m = 76$) and LA ($n = 3725$, $m = 169$). For each rape crime we calculate the reporting delay in days as the difference between the reporting date and the occurrence date, $\Delta t_i = t_i^{\text{Reporting}} - t_i^{\text{Occurrence}}$. As the distribution of reporting delays is characterized by a heavy tail (figure 1), we further process the data to make it suitable for predictive modelling. We use the reporting delays to create three separate binary indicators $d_i^{\{\text{day,week,month}\}}$ of whether a rape event was reported within one day, one week or one month, respectively,

$$d_i^{\text{day}} = \begin{Bmatrix} 1 & \text{for } \Delta t_i \leq 1 \\ 0 & \text{for } \Delta t_i \geq 2 \end{Bmatrix}, \quad d_i^{\text{week}} = \begin{Bmatrix} 1 & \text{for } \Delta t_i \leq 7 \\ 0 & \text{for } \Delta t_i \geq 8 \end{Bmatrix}, \quad d_i^{\text{month}} = \begin{Bmatrix} 1 & \text{for } \Delta t_i \leq 30 \\ 0 & \text{for } \Delta t_i \geq 31 \end{Bmatrix}.$$

The decision of using one day, one week and one month as thresholds, respectively, does not only stem from their heuristic value, but is also motivated by insights from forensics and mental health research. Survivors reporting and seeking support within 1 day have been shown to have better treatment outcomes [29], while their medical examination can provide the best possible forensic evidence [30] to

assist prosecution. For female survivors of sexual assault with penetration, traces of DNA evidence last for about one week due to vaginal drainage, menstruation and sperm degradation [30]. Lastly, rape reported one month or longer after the occurrence is associated with a severe increase in psychological distress for survivors [31]. One month is also the approximate period to allow for the recovery of forensic evidence of potential drugging from the survivor's hair [32].

The binary indicators $d_i^{\{day,week,month\}}$ serve as the dependent variables for an event-level classification problem. Beyond the binary indicators, we also test modelling approaches using a log-transformation on the reporting delays, $\log(\Delta t_i)$, which constitutes a regression problem.

To assess potential variation between policing districts, we also aggregate the reporting delays geographically: we chose the NYPD police precincts and the LAPD basic car districts for aggregation, both referred to as 'police districts' in this study. Each rape event is recorded in a police district, which allows us to easily aggregate reporting delays at this level. In this way, we can compute the *proportions p* of rapes in each area which are reported within one day, one week or one month, respectively,

$$p_j^{day} = \frac{1}{n_j} \sum_{i \in j} d_i^{day}, \quad p_j^{week} = \frac{1}{n_j} \sum_{i \in j} d_i^{week}, \quad p_j^{month} = \frac{1}{n_j} \sum_{i \in j} d_i^{month}.$$

Note that we will only use the proportions $p_j$ for assessing spatial variation in reporting delays, and not for predictive modelling. This is done so we can assess spatial variation both within and between police districts. Figure 1 (for the city of LA) gives a first impression of the spatial variation in $d_i^{\{day,week,month\}}$ and $p_j^{\{day,week,month\}}$.

We now consider our predictor variables. We first access information about each rape event separately. This includes information about the survivor (age, sex, race), the crime date (weekend, holiday) and the crime location. Importantly, while LAPD reports crime locations (as point coordinates) at the block level, NYPD reports the location of rape crimes only at the police district level. This is done to obfuscate the exact crime location in order to protect survivor identities. Beyond crime-specific information, we access socio-demographic and economic data on a census tract level for the operational areas of NYPD and LAPD. For the Los Angeles data, we assign each rape crime event socio-demographic and economic characteristics from the census tract within which the crime event was recorded. For the New York data, we must aggregate at the police district level, noting that each police district may intersect with multiple census tracts. For this process, we use a population-weighted aggregation approach: following the DE-9IM formulation [33], we denote a set of census tract polygons $b_k$ which hold area-specific socio-demographic and economic information and which intersect with police district polygons $a_j$. We first compute the set of intersection polygons between census tracts and police districts as $II_{k \cap j} = I(b_k) \cap I(a_j)$, where $I$ represents the interior of a polygon. We then compute the proportion of each census tract area $b_k$ which intersects with the police district $a_j$ as $P_{j|k}^A = A(II_{k \cap j})/A(b_k)$, so that $\sum_j P_{j|k}^A = 1$. We denote the population in a census tract $b_k$ as $x_k$. Assuming a spatially uniformly distributed population in a census tract, we estimate the population living in $II_{k \cap j}$ as $x_{k \cap j} = P_{j|k}^A x_k$. Hence we can calculate the area-weighted population in a police district $a_j$ as $x_j = \sum_k x_{k \cap j}$ and the population proportion for census tract $b_k$ in a police district $a_j$ as $P_{k|j}^x = x_{k \cap j}/x_j$. We use these population proportions to weight all predictor variables during the aggregation process. The predictors aggregated in this way include demographic information (population age, sex, race), housing information (average room occupancy, household size), economic information (median income, unemployment rate) and socio-cultural information (educational attainment, language spoken at home). For a full list of the predictors, please see table 1.

## 2.2. Spatial autocorrelation

To account for potential spatial dependencies in our data, we examine the spatial autocorrelation of rape reporting delays, and the RSA of our models' residuals, throughout the study. We use both the global and local Moran's I test [34,35]. These statistics capture local correlation structures in a multi-dimensional space and help us to explore whether rape reporting delays exhibit spatial dependencies, which could be exploited in a predictive modelling framework. The local Moran's I statistic is defined as follows:

$$I_j = (m-1)\frac{p_j - \bar{p}}{\sum_{l=1,l \neq j}^m (p_l - \bar{p})^2} \sum_{l=1,l \neq j}^m w_{j,l}(p_l - \bar{p})$$

where $\bar{p}$ gives the mean of $p_j$ and $w_{j,l}$ indicates a neighbourhood weight between areas $j$ and $l$. We create a binary spatial weight matrix using a 'queen' neighbourhood definition. That is, we assign $w_{j,l} = 1$ when the area polygons $j$ and $l$ touch, and $w_{j,l} = 0$ otherwise. For the event-level data, the formula is identical, except

**Table 1.** Overview of the set of predictor variables used in our study. The values stem from either the crime report directly, or the American Community Survey (ACS). Survivor characteristics and temporal factors are only available in the event-level, disaggregated model. The demographic and economic features are collected from the most granular available geographical area, the census tract or aggregated on police precinct level (see §2.1). For New York, the spatial location is given as the centroid of the respective police precinct, while for Los Angeles we have block-level geographical coordinates.

| name | description |
| --- | --- |
| demographic and economic | |
| 'pop.total' | total population in area |
| 'pop.male' | % of population: male |
| 'pop.u20' | % of population: age <20 |
| 'pop.20.64' | % of population: age 20–64 |
| 'pop.o64' | % of population: age >64 |
| 'pop.married' | % of population: married |
| 'pop.white' | % of population: white |
| 'pop.black' | % of population: black |
| 'pop.native' | % of population: native |
| 'pop.asian' | % of population: asian |
| 'pop.pacific' | % of population: pacific |
| 'pop.latino' | % of population: latino |
| 'occupants.1.5' | % of occupied housing units with >1.5 occupants per room |
| 'rent.35.income' | % of population: rent constitutes >35% of income |
| 'occ.units' | total occupied housing units |
| 'pop.single.hh' | % of population: living in single household |
| 'hh.size' | average household size |
| 'no.school' | % of population: no high-school degree |
| 'poverty' | % of population: living in poverty |
| 'med.income' | median income |
| 'pop.no.insur' | % of population: no insurance |
| 'pop.unempl' | % of population: unemployed |
| 'lan.english' | % of population: main language at home English |
| 'lan.spanish' | % of population: main language at home Spanish |
| 'lan.yiddish' | % of population: main language at home Yiddish |
| 'lan.russian' | % of population: main language at home Russian |
| 'lan.polish' | % of population: main language at home Polish |
| 'lan.armenian' | % of population: main language at home Armenian |
| 'lan.hindi' | % of population: main language at home Hindi |
| 'lan.chinese' | % of population: main language at home Chinese |
| 'lan.vietnamese' | % of population: main language at home Vietnamese |
| 'lan.arabic' | % of population: main language at home Arabic |
| 'lan.hebrew' | % of population: main language at home Hebrew |
| 'lan.african' | % of population: main language at home African |
| spatial | |
| 'lon' | crime location (longitude) |
| 'lat' | crime location (latitude) |

(*Continued.*)

**Table 1.** (Continued.)

| name | description |
| --- | --- |
| **temporal** | |
| 'winter' | crime occurrence date: winter |
| 'spring' | crime occurrence date: spring |
| 'summer' | crime occurrence date: summer |
| 'fall' | crime occurrence date: fall |
| 'weekend' | crime occurrence date: weekend |
| 'federal' | crime occurrence date: federal holiday |
| 'christian' | crime occurrence date: Christian holiday |
| 'muslim' | crime occurrence date: Muslim holiday |
| 'jewish' | crime occurrence date: Jewish holiday |
| 'hindu' | crime occurrence date: Hindu holiday |
| 'celebration' | crime occurrence date: popular celebration |
| **survivor characteristics** | |
| 'vict.white' | survivor ethnicity: white |
| 'vict.black' | survivor ethnicity: black |
| 'vict.latin' | survivor ethnicity: latino |
| 'vict.asian' | survivor ethnicity: asian |
| 'vict.female' | survivor sex: female |
| 'vict.u18' | survivor age: <18 years |
| 'vict.18.24' | survivor age: 18–24 years |
| 'vict.25.44' | survivor age: >24–44 years |
| 'vict.45.64' | survivor age: 45–64 years |
| 'vict.o64' | survivor age: >64 years |

that the spatial weights matrix is calculated using a k-nearest-neighbour (kNN) approach where the k spatially closest events to each observation are treated as neighbours.

The global Moran's I statistic can be defined as

$$I_{\text{global}} = \frac{m}{\sum_{j=1}^{m} \sum_{l=1}^{m} w_{j,l}} \frac{\sum_{j=1}^{m} \sum_{l=1}^{m} w_{j,l}(p_j - \bar{p})(p_l - \bar{p})}{\sum_{j=1}^{m}(p_j - \bar{p})^2}.$$

Both the local and the global Moran's I statistics come with corresponding pseudo $p$-values, based on a permutation test, which indicate statistical significance of the local and global spatial correlation structure, respectively.

## 2.3. Modelling framework

The objective of our modelling approach is to predict reporting delays of rape crime. As outlined above, this is treated both as a classification task, where we seek to predict whether a rape crime was reported within a certain time period (one day, one week, one month), and a regression task, where we seek to predict the log-transformed reporting delay. Building the predictive models relies on an extensive set of predictor variables, including many never used before in the context of rape reporting delays. An overview of these features can be found in table 1. Our modelling pipeline is illustrated in figure 2.

Our models seek to predict the binary, event-level classes in $d_i^{\{\text{day,week,month}\}}$ or log-transformed reporting delays ($\log(\Delta t_i)$) using sets of socio-economic and demographic predictors $X_i$ (stemming from either the census tract (Los Angeles) or the police precinct (New York) the crime was committed in), crime location coordinates $C_i$ (police precinct centroids in New York), police district dummies $Q_i$, temporal predictors $Z_i$ and survivor information predictors $V_i$ so that $[d_i, \log(\Delta t_i)] \sim f(X_i, C_i, Q_i, Z_i, V_i)$. For these tasks we compare penalized linear models (LMs; linear regression and logistic regression), random forest (RF) [36] and

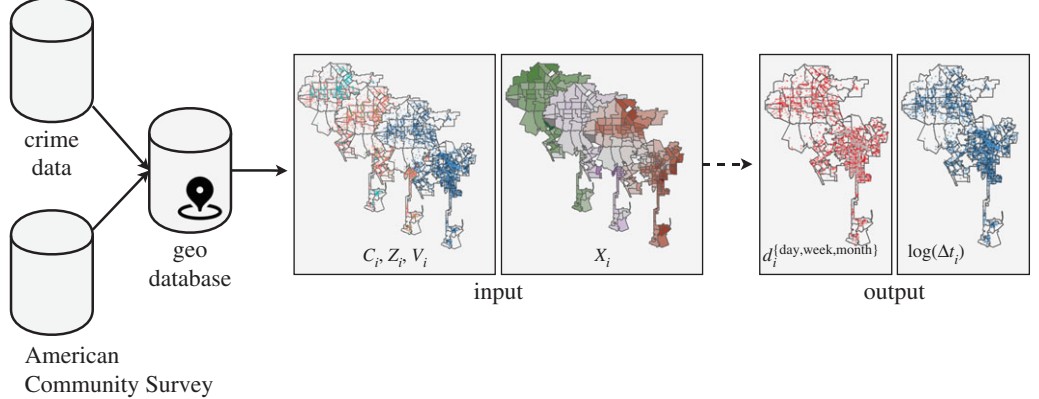

**Figure 2.** Data processing pipeline and modelling framework for the conducted experiments.

gradient boosting (GB) [37] algorithms. For models including spatial coordinates $C_i$ we seek to learn spatial dependencies and opt for Gaussian process (GP) models. These are kernel-based models and as such especially capable of handling autoregressive data structures, as well as guaranteeing Gaussian residuals. To get deeper insight into the predictive models and the importance of the different feature categories, we build our models step-by-step, starting with our non-spatial baseline models, followed by models incorporating spatial context in different ways and concluding with our final models, combining all available predictors.

### 2.3.1. Non-spatial baseline models

**(NS1)** *Survivor characteristics:* Our first baseline model solely seeks to predict reporting delays using only survivor characteristics such as age and ethnicity: $[d_i, \log(\Delta t_i)] \sim f(V_i)$.

　　**(NS2)** *Survivor + temporal characteristics:* Our second baseline model adds temporal features of the day the crime was committed, e.g. whether that day was a national or religious holiday or on a weekend: $[d_i, \log(\Delta t_i)] \sim f(Z_i, V_i)$.

### 2.3.2. Spatial models

**(S1)** *Random effects model:* The random effects model expands the predictor variables by taking the socio-economic and demographic features of the area the crime was committed in into account: $[d_i, \log(\Delta t_i)] \sim f(X_i, Z_i, V_i)$.

　　**(S2)** *Fixed effects model:* The fixed effects model instead uses dummies for the police district a crime was committed in to account for spatial effects: $[d_i, \log(\Delta t_i)] \sim f(Q_i, Z_i, V_i)$.

　　**(S3)** *Combined spatial model:* Our final model uses all available predictors except the police district dummies: $[d_i, \log(\Delta t_i)] \sim f(X_i, C_i, Z_i, V_i)$.

　　In order to deal with the high-dimensional feature space, we use feature selection techniques in all our models. RFs and GB both have built-in feature selection. For our LMs, we apply L1 (Lasso) regularization, allowing unimportant model coefficients to be reduced to zero which eliminates the respective predictor. For the GPs, we apply pre-training feature selection, again using L1 linear regularizers. For model S3, point coordinates are excluded from the feature selection process and added as predictors by default. Note that we only use GPs for model S3, which explicitly includes spatial coordinates. Linear models, RF and GB are all implemented in R (v. 4.0.2) using the *caret* machine learning framework and the model-specific *glmnet*, *randomForest* and *xgboost* packages. GP models are implemented using the efficient *GPyTorch* framework in Python 3.

## 3. Results

### 3.1. Summary statistics

#### 3.1.1. Examining reporting delays

Many survivors of rape only report the crime to police after substantial delays, which can span days, months or years. This leads to heavy-tailed distributions with high mean reporting delays. Therefore,

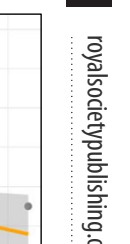

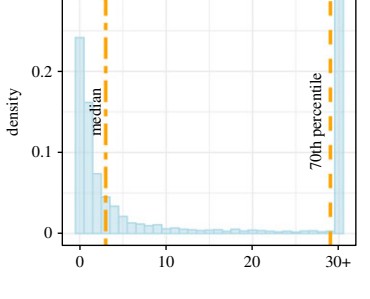
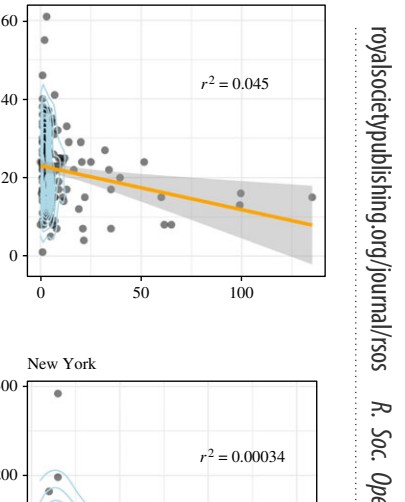
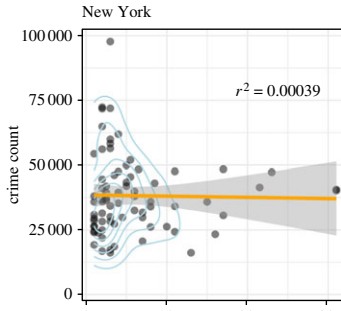
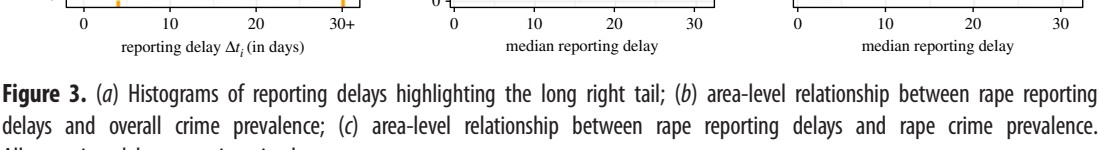

**Figure 3.** (*a*) Histograms of reporting delays highlighting the long right tail; (*b*) area-level relationship between rape reporting delays and overall crime prevalence; (*c*) area-level relationship between rape reporting delays and rape crime prevalence. All reporting delays are given in days.

investigating median reporting delays, log-transformed reporting delays or the proportion of rapes that exceed a certain reporting delay threshold can be more informative. In our observed data, the median reporting delay for Los Angeles is 3 days, and that of New York is 4 days. However, approximately 30% of all recorded rapes are reported one month or later after the crime was committed, as figure 3*a* highlights. Rape, like assault or robbery, has an immediate effect on the survivor: this contrasts with theft for instance, which might only be noted some time after the crime was committed. This implies that, unless the survivor is physically incapacitated by the attack, they theoretically could report the crime immediately. However, when comparing reporting delays of assault to those of rape, we see a stark difference: assault is reported much earlier, usually on the day of occurrence.

What factors can explain these observed delays in the reporting of rape crimes? We begin the analysis by looking into the relationship between rape reporting delays and general crime prevalence. For this, we examine the median reporting delay for each police district in Los Angeles and New York and compare it with respective overall crime counts (figure 3*b*) and rape crime counts (figure 3*c*) in that district. In both cities, there is little association between rape reporting delays and criminal activity; Los Angeles shows a weak negative correlation ($R^2 = 0.05$) between the two, and New York shows no correlation. Further, we compare reporting delays to the prevalence of rape crime specifically. Here, we observe the same patterns as for overall crime counts, with Los Angeles exhibiting a weak negative correlation between rape counts and reporting delays. Lastly, we find no correlation between rape reporting delays and reporting delays on non-rape crime in either city. The median reporting delays for non-rape crime are very similar across police precincts and cities, always falling between 0- and 1-day delays. From this, we conclude that the process for rape reporting is distinct from general patterns of criminal activity and crime reporting. This observation also serves as one of our main motivations for a thorough analysis of what predicts rape reporting delays.

### 3.1.2. Spatial patterns in reporting behaviour

Next, we examine the process of rape reporting over time and space. First, we seek to determine if rape reporting delays exhibit autocorrelation structures over space. We apply Moran's I tests for statistical significance of global and local spatial autocorrelation (see the previous section). For our initial data assessment, we examine spatial autocorrelation in area-level reporting delay proportions $p_i^{\{day,week,month\}}$

**Table 2.** p-values of global Moran's I tests for spatial autocorrelation of the proportions $p_j^{\{day,week,month\}}$ and the log-transformed reporting delay $\log(\Delta t_i)$ for LA and NY. To create the spatial neighbourhood matrix required for computing Moran's I, we use k-nearest-neighbours with $k = 50$ for NY and all observations in the same police district as neighbours in LA. Significance codes: 0.05\*, 0.01\*\*, 0.001\*\*\*.

| LA | | | | NY | | | |
|---|---|---|---|---|---|---|---|
| $p_j^{day}$ | $p_j^{week}$ | $p_j^{month}$ | $\log(\Delta t_i)$ | $p_j^{day}$ | $p_j^{week}$ | $p_j^{month}$ | $\log(\Delta t_i)$ |
| 0.067 | 0.009\*\* | 0.002\*\* | 0.000\*\*\* | 0.013\* | 0.01\*\* | 0.006\*\* | 0.000\*\*\* |

and log-transformed reporting delays $\log(\Delta t_i)$ for both cities. At the area level, we find evidence for global spatial autocorrelation in both cities, as well as areas in both cities that exhibit significant local correlation structures. At the event level, we also find evidence for significant global and local spatial autocorrelation in both cities. However, note that only Los Angeles data comes with granular spatial coordinates. Notably, the local spatial autocorrelation patterns seem to intersect relatively well between area- and event-level data, demonstrating substantial correlation in reporting delays between nearby areas. Table 2 provides the global Moran's I p-values for tests on New York and Los Angeles data.

We also apply a spatial permutation test to examine the distribution of the median reporting delays for each police district, in order to determine whether there is more variation in median delay across districts than one would expect by chance, indicating that cases with high reporting delays tend to cluster within a district. To conduct the permutation test, we randomly assign each rape crime to a police district, keeping the overall number of rape crimes in each district fixed to the original number. We then compute the median reporting delay for each district and rank the districts by this measure. This process is repeated 1000 times, and then we compute the mean and 95% confidence interval of median delays for each rank. These are then compared with the original ranked median reporting delays. In this way, we can assess if a random spatial assignment of cases would yield the same distribution of median reporting delays as the true data. We show the results for this permutation test in figure 4a, highlighting that, particularly in Los Angeles, there is much more spatial variation in the median reporting delay than one would expect by chance. Lastly, we run one-sided tests for spatial disparities in median reporting time across districts, using the permutation data. For New York, we find that the median reporting delays have a Gini coefficient of 0.497 ($p = 0.046$) and an inter-decile range of 13.0 ($p = 0.048$), indicating significant disparities. For Los Angeles, the Gini coefficient is 0.554 ($p = 0.076$) and the inter-decile range is 19.1 ($p = 0.002$), also indicating substantial and unexplained spatial variation in rape reporting delays. Our findings suggest that the local area in which a rape occurs is a critical factor contributing to delayed rape reporting. This raises the question of whether the observed spatial patterns can be attributed to a purely spatial process, or whether they are actually the manifestations of underlying social processes with spatial dependencies (such as segregation or low/high-income areas). We examine this question further in the subsection on predictive modelling below.

### 3.1.3. Temporal patterns in reporting behaviour

Before this, however, we also investigate temporal dependencies of reporting delays; that is, we seek to assess whether rapes that were committed on certain days, or during certain weeks or months exhibit higher or lower reporting delays. For this analysis, it is important to note that the reported occurrence date is not always the true occurrence date: survivors who report rape after a long delay, or who are severely traumatized by the attack, are not always able to recall the exact date of the event. In these cases, both observed police departments follow a similar approach; the occurrence date is given as the closest possible approximation. For instance, if a survivor recalls the rape to have occurred in 'early March', 1 March will be recorded as the data of occurrence. This procedure leads to substantial spikes in the median reporting delays on the first and middle days of each month. However, some spikes appear unrelated to this practice. We, therefore, inspect the time of rape occurrence, specifically assessing in which season the crime happened, the day (including if this day was a weekend), and whether that day was a federal holiday or religious festival. While weekends appear unrelated to reporting delays, we do find substantial spikes in median reporting delays on some federal holidays and widely celebrated festivals. Figure 4b highlights observed spikes on Valentine's Day, Halloween and Thanksgiving. This suggests that the day a rape happened might also have some predictive power with regard to its

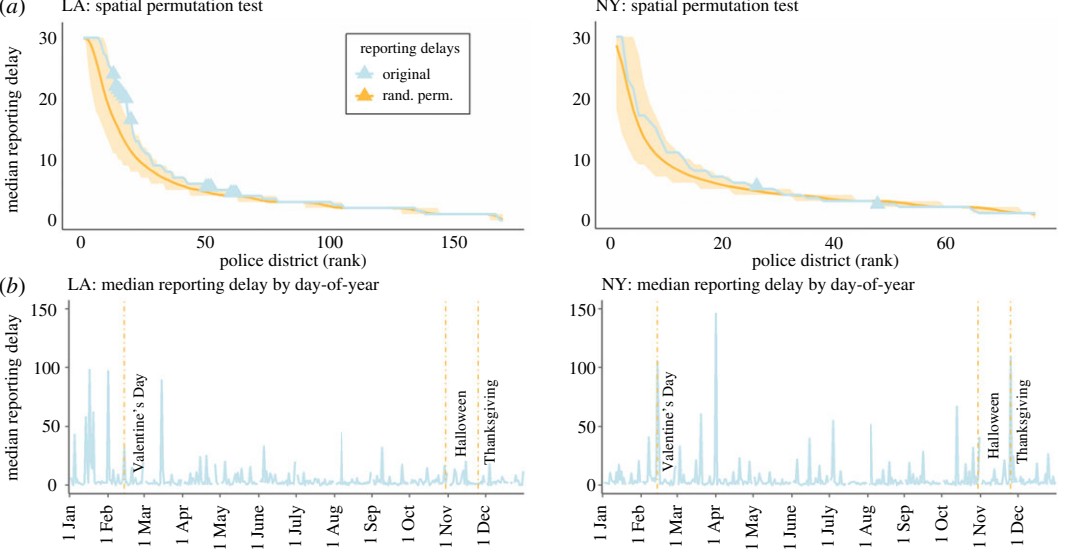

**Figure 4.** Spatial and temporal dependencies of rape reporting delays. (*a*) Spatial permutation test of median reporting delays (95% confidence interval under the null hypothesis of spatial randomness) compared with true median delays by police district. Ranked observations outside the 95% CI for that rank are marked with triangles. (*b*) Median rape reporting delays by day of occurrence (aggregated over all observed years), with certain holidays highlighted by dashed lines.

respective reporting delay. Therefore, we conclude that rape reporting delays are in fact not independent and identically distributed (iid) across space and time, but rather exhibit substantial variation depending on both spatial and temporal factors.

## 3.2. Forecasting reporting delays

### 3.2.1. Predictive performance

We now move on to the main component of our study, the predictive modelling framework for rape reporting delays. We report results from the five predictive models (NS1–NS2, S1–S3) and the four statistical and machine learning methods (LMs, RFs, GB and GPs) outlined in §2.3. To minimize the risk of overfitting and to provide robust outcomes, we report findings from 10-fold cross-validation.

For classification tasks, we report the out-of-sample accuracy and area under the receiver operating characteristic curve (AUC) metrics. For accuracy scores, we also report a naive 'majority vote' baseline, where accuracy is calculated by assigning all observations the majority label. For regression tasks on log-transformed reporting delays, we report residual mean squared error (RMSE) and $R$-squared ($R^2$) metrics. The results for all models and algorithms can be found in tables 3–10, and are discussed in detail below. We can observe that the different prediction algorithms perform similarly, with LMs and GPs typically performing best. Our results show a large improvement over an uninformed guess and can hence provide a basis for both policy (to mitigate reporting delays for vulnerable areas and subpopulations and to support survivors) as well as further predictive analyses.

Tables 3 and 4 provide classification scores for models predicting $d_i^{\text{day}}$. Here, we obtain the best AUC scores of 0.688 and 0.643, for New York and Los Angeles, respectively, from the S3 model.

Tables 5 and 6 provide classification scores for models predicting $d_i^{\text{week}}$. We can see an improvement over the $d_i^{\text{day}}$ models, obtaining the best AUC scores of 0.701 and 0.692, for New York and Los Angeles, respectively, from the S3 and the S1 model.

Tables 7 and 8 provide classification scores for models predicting $d_i^{\text{month}}$. We can again see an improvement and our best overall classification scores here: model S3 provides AUC scores of 0.710 for NY and 0.718 for LA.

Lastly, tables 9 and 10 provide regression scores for models predicting $\log(\Delta t_i)$. We can observe that the model S3 substantially improves predictive performance over the baseline models. We record RMSE values of 0.202 for NY and 0.274 for LA. The performance increase for LA is particularly noteworthy, as it highlights the high predictive value of the spatial coordinates of a crime.

**Table 3.** Accuracy of different algorithms across five spatial and non-spatial models with outcome variable $d_i^{\text{day}}$. For comparison, naive prediction of the majority class would achieve accuracy of 0.588 for New York and 0.597 for Los Angeles. The respectively best performing learning algorithm (LM, RF, GB, GP) is highlighted in bold.

|     | New York | | | | | Los Angeles | | | | |
| --- | --- | --- | --- | --- | --- | --- | --- | --- | --- | --- |
|     | NS1 | NS2 | S1 | S2 | S3 | NS1 | NS2 | S1 | S2 | S3 |
| LM  | 0.634 | 0.639 | 0.639 | 0.641 | 0.641 | 0.611 | 0.605 | 0.610 | 0.596 | 0.599 |
| RF  | **0.646** | 0.641 | 0.606 | 0.615 | 0.595 | 0.612 | 0.611 | 0.583 | 0.578 | 0.584 |
| GB  | 0.645 | 0.638 | 0.616 | 0.632 | 0.616 | 0.611 | 0.604 | 0.575 | 0.580 | 0.570 |
| GP  | — | — | — | — | **0.646** | — | — | — | — | **0.620** |

**Table 4.** AUC of different algorithms across five spatial and non-spatial models with outcome variable $d_i^{\text{day}}$. The respectively best performing learning algorithm (LM, RF, GB, GP) is highlighted in bold.

|     | New York | | | | | Los Angeles | | | | |
| --- | --- | --- | --- | --- | --- | --- | --- | --- | --- | --- |
|     | NS1 | NS2 | S1 | S2 | S3 | NS1 | NS2 | S1 | S2 | S3 |
| LM  | 0.660 | 0.680 | 0.680 | 0.680 | 0.680 | 0.620 | 0.630 | 0.640 | 0.630 | 0.640 |
| RF  | 0.670 | 0.670 | 0.650 | 0.660 | 0.630 | 0.600 | 0.630 | 0.610 | 0.610 | 0.610 |
| GB  | 0.670 | 0.670 | 0.640 | 0.660 | 0.650 | 0.620 | 0.620 | 0.590 | 0.610 | 0.590 |
| GP  | — | — | — | — | **0.688** | — | — | — | — | **0.643** |

**Table 5.** Accuracy of different algorithms across five spatial and non-spatial models with outcome variable $d_i^{\text{week}}$. For comparison, naive prediction of the majority class would achieve accuracy of 0.56 for New York and 0.603 for Los Angeles. The respectively best performing learning algorithm (LM, RF, GB, GP) is highlighted in bold.

|     | New York | | | | | Los Angeles | | | | |
| --- | --- | --- | --- | --- | --- | --- | --- | --- | --- | --- |
|     | NS1 | NS2 | S1 | S2 | S3 | NS1 | NS2 | S1 | S2 | S3 |
| LM  | 0.671 | 0.671 | **0.672** | **0.672** | 0.669 | 0.676 | 0.679 | **0.681** | 0.670 | 0.669 |
| RF  | 0.670 | 0.668 | 0.625 | 0.618 | 0.599 | 0.676 | 0.665 | 0.630 | 0.616 | 0.612 |
| GB  | 0.668 | 0.663 | 0.628 | 0.655 | 0.626 | 0.676 | 0.661 | 0.612 | 0.647 | 0.619 |
| GP  | — | — | — | — | 0.667 | — | — | — | — | 0.676 |

**Table 6.** AUC of different algorithms across five spatial and non-spatial models with outcome variable $d_i^{\text{week}}$. The respectively best performing learning algorithm (LM, RF, GB, GP) is highlighted in bold.

|     | New York | | | | | Los Angeles | | | | |
| --- | --- | --- | --- | --- | --- | --- | --- | --- | --- | --- |
|     | NS1 | NS2 | S1 | S2 | S3 | NS1 | NS2 | S1 | S2 | S3 |
| LM  | 0.690 | 0.700 | 0.700 | 0.700 | 0.700 | 0.670 | 0.670 | **0.690** | 0.670 | 0.670 |
| RF  | 0.690 | 0.700 | 0.670 | 0.680 | 0.660 | 0.660 | 0.680 | 0.660 | 0.630 | 0.630 |
| GB  | 0.690 | 0.680 | 0.670 | 0.680 | 0.660 | 0.670 | 0.670 | 0.620 | 0.650 | 0.630 |
| GP  | — | — | — | — | **0.701** | — | — | — | — | 0.682 |

For the non-spatial models, we can observe that adding information on the occurrence date of a crime (NS2) helps to increase performance compared with NS1, highlighting the predictive value of temporal features. Our step-by-step modelling approach also allows for a deeper understanding of the observed spatial patterns in reporting delays. The results show that the non-spatial baseline models NS1 and NS2 are consistently outperformed by the spatial models, particularly the combined model S3. We can also see that the random effects spatial model S1 usually performs better than the fixed effects

**Table 7.** Accuracy of different algorithms across five spatial and non-spatial models with outcome variable $d_i^{month}$. For comparison, naive prediction of the majority class would achieve accuracy of 0.646 for New York and 0.702 for Los Angeles. The respectively best performing learning algorithm (LM, RF, GB, GP) is highlighted in bold.

|  | New York | | | | | Los Angeles | | | | |
|---|---|---|---|---|---|---|---|---|---|---|
|  | NS1 | NS2 | S1 | S2 | S3 | NS1 | NS2 | S1 | S2 | S3 |
| LM | 0.678 | 0.688 | 0.686 | 0.687 | 0.685 | 0.727 | 0.738 | 0.738 | 0.731 | 0.729 |
| RF | 0.680 | 0.679 | 0.656 | 0.665 | 0.649 | 0.729 | 0.733 | 0.707 | 0.698 | 0.702 |
| GB | 0.681 | 0.678 | 0.654 | 0.667 | 0.655 | 0.729 | 0.732 | 0.696 | 0.722 | 0.698 |
| GP | — | — | — | — | **0.690** | — | — | — | — | **0.742** |

**Table 8.** AUC of different algorithms across five spatial and non-spatial models with outcome variable $d_i^{month}$. The respectively best performing learning algorithm (LM, RF, GB, GP) is highlighted in bold.

|  | New York | | | | | Los Angeles | | | | |
|---|---|---|---|---|---|---|---|---|---|---|
|  | NS1 | NS2 | S1 | S2 | S3 | NS1 | NS2 | S1 | S2 | S3 |
| LM | 0.680 | 0.700 | **0.710** | 0.700 | **0.710** | 0.690 | 0.710 | 0.710 | 0.700 | 0.700 |
| RF | 0.670 | 0.700 | 0.670 | 0.670 | 0.650 | 0.660 | 0.700 | 0.690 | 0.690 | 0.670 |
| GB | 0.680 | 0.690 | 0.670 | 0.690 | 0.680 | 0.680 | 0.700 | 0.670 | 0.690 | 0.670 |
| GP | — | — | — | — | **0.710** | — | — | — | — | **0.718** |

**Table 9.** RMSE of different algorithms across five spatial and non-spatial models with outcome variable $\log(\Delta t_i)$. The respectively best performing learning algorithm (LM, RF, GB, GP) is highlighted in bold.

|  | New York | | | | | Los Angeles | | | | |
|---|---|---|---|---|---|---|---|---|---|---|
|  | NS1 | NS2 | S1 | S2 | S3 | NS1 | NS2 | S1 | S2 | S3 |
| LM | 0.205 | 0.203 | 0.203 | 0.203 | 0.203 | 0.285 | 0.278 | 0.278 | 0.281 | 0.281 |
| RF | 0.205 | 0.205 | 0.212 | 0.211 | 0.215 | 0.285 | 0.280 | 0.289 | 0.297 | 0.293 |
| GB | 0.205 | 0.208 | 0.222 | 0.211 | 0.223 | 0.286 | 0.287 | 0.318 | 0.294 | 0.309 |
| GP | — | — | — | — | **0.202** | — | — | — | — | **0.274** |

**Table 10.** $R^2$ of different algorithms across five spatial and non-spatial models with outcome variable $\log(\Delta t_i)$. The respectively best performing learning algorithm (LM, RF, GB, GP) is highlighted in bold.

|  | New York | | | | | Los Angeles | | | | |
|---|---|---|---|---|---|---|---|---|---|---|
|  | NS1 | NS2 | S1 | S2 | S3 | NS1 | NS2 | S1 | S2 | S3 |
| LM | 0.097 | 0.117 | 0.119 | 0.118 | **0.120** | 0.140 | 0.186 | 0.187 | 0.172 | 0.170 |
| RF | 0.100 | 0.112 | 0.082 | 0.097 | 0.083 | 0.139 | 0.182 | 0.142 | 0.153 | 0.146 |
| GB | 0.100 | 0.095 | 0.065 | 0.081 | 0.065 | 0.135 | 0.157 | 0.094 | 0.149 | 0.109 |
| GP | — | — | — | — | 0.119 | — | — | — | — | **0.202** |

model S2. While this speaks to the general value of implicit and explicit spatial features for explaining variance in the distribution of reporting delays, the findings suggest that socio-economic and demographic factors, rather than other unmodelled features of a given police district, are important for forecasting reporting delays.

Next, we revisit the question posed in §3.1.2: can the observed spatial autocorrelation in reporting delays be attributed to socio-economic and demographic characteristics? If that hypothesis is true, we

**Table 11.** Residual spatial autocorrelation of the best performing models for NY. Provided are $p$-values from global Moran's I tests for spatial autocorrelation, with significant $p$-values ($p < 0.05$) in bold. To create the spatial neighbourhood matrix required for computing Moran's I, we define all crimes happening in the same police district as neighbours.

| | NS1 | NS2 | S1 | S2 | S3 |
|---|---|---|---|---|---|
| $d_i^{day}$ | 0.081 | 0.108 | 0.704 | 0.125 | 1.000 |
| $d_i^{week}$ | **0.005** | **0.019** | 0.763 | **0.027** | 1.000 |
| $d_i^{month}$ | **0.009** | 0.079 | 0.785 | 0.083 | 1.000 |
| $\log(\Delta t_i)$ | **0.001** | **0.006** | 0.636 | **0.008** | 1.000 |

**Table 12.** Residual spatial autocorrelation of the best performing models for LA. Provided are $p$-values from global Moran's I tests for spatial autocorrelation, with significant $p$-values ($p < 0.05$) in bold. To create the spatial neighbourhood matrix required for computing Moran's I, we use k-nearest-neighbours with $k = 50$.

| | NS1 | NS2 | S1 | S2 | S3 |
|---|---|---|---|---|---|
| $d_i^{day}$ | 0.100 | **0.028** | 0.369 | 0.369 | 1.000 |
| $d_i^{week}$ | 0.078 | 0.056 | 0.699 | 0.699 | 1.000 |
| $d_i^{month}$ | **0.002** | **0.002** | 0.125 | 0.125 | 1.000 |
| $\log(\Delta t_i)$ | **0.001** | **0.000** | 0.142 | 0.979 | 1.000 |

should observe the spatially uncorrelated model residuals. Thus, we test for RSA in our models, that is whether our model error term is itself spatially autocorrelated. We test residuals of the best performing method for each city (LA, NY), model (NS1, NS2, S1, S2, S3) and outcome variable ($[d_i, \log(\Delta t_i)]$) for spatial autocorrelation, using the global Moran's I test. These results are presented in tables 11 and 12.

The tests for RSA confirm that both non-spatial baseline models (NS1 and NS2) exhibit spatially autocorrelated error terms. While non-spatial features such as survivor demographics, seasonality and holidays explain a substantial fraction of the variation in rape reporting delays, significant spatial autocorrelation remains after accounting for these features. While this would be a major issue for causal identification (since the independence assumption for LM residuals is violated), it is not necessarily a problem in a purely prediction setting, as is our focus here. However, it suggests that accounting for spatial dependencies might help increase predictive performance—which is exactly what we observe in our models. Either controlling for population demographics and socio-economics of the local area (S1, S3), or explicitly modelling the spatial dimension (S2, S3), reduces or removes the remaining spatial autocorrelation and improves predictive accuracy. Out of all spatial models, only S2 for New York shows some significant RSA ($p < 0.05$). All S1 and S3 models account for the spatial correlation in the output variable, leaving uncorrelated residuals. Finally, our findings also show the limitations of our study, highlighting how noisy publicly available data on rape reporting is: even with a large set of predictors and a carefully designed modelling framework, we are only able to disentangle a portion of the observed variation in reporting delays.

### 3.2.2. Assessing feature importance

As our final analysis step, we assess the model coefficients of our LMs S3. We report significant coefficients ($p < 0.05$) for logistic regression models ($d_i^{\{day,week,month\}}$) and LMs ($\log(\Delta t_i)$), estimated using ordinary least squares linear regression, in table 13 (LA) and table 14 (NY). We did not observe sufficiently strong correlations across predictors to cause issues with multicollinearity. The significant predictors display high consistency across the four dependent variables ($d_i^{\{day,week,month\}}$ and $\log(\Delta t_i)$) and the two cities (NY and LA) under consideration. In particular, we observe longer reporting delays for rape survivors who were under the age of 18 or non-female, longer reporting delays for Asian or Latino survivors as compared with Black or White survivors, and longer reporting delays for rape crimes committed in the winter or on federal holidays. For some of our models, uninsured populations and poverty in the local area corresponded to longer reporting delays, while a higher

**Table 13.** Model coefficients and p-values of significant ($p < 0.05$) predictors taken from linear models using $d_i^{(day,week,month)}$ and $\log(\Delta t_i)$ as dependent variables and using data from LA.

**LA**

| dep. var. | $d_i^{day}$ | | $d_i^{week}$ | | | $d_i^{month}$ | | | $\log(\Delta t_i)$ | | |
|---|---|---|---|---|---|---|---|---|---|---|---|
| pred. var. | coeff. | p-val. | pred. var. | coeff. | p-val. | pred. var. | coeff. | p-val. | pred. var. | coeff. | p-val. |
| vict.u18 | −1.683 | 0.000 | federal | −1.088 | 0.000 | federal | −1.241 | 0.000 | federal | 1.876 | 0.000 |
| federal | −0.627 | 0.000 | vict.u18 | −2.241 | 0.000 | vict.u18 | −1.908 | 0.000 | vict.u18 | 2.153 | 0.000 |
| christian | −0.403 | 0.003 | vict.female | 1.175 | 0.002 | winter | −0.412 | 0.000 | winter | 0.459 | 0.000 |
| vict.black | 0.486 | 0.007 | vict.black | 0.552 | 0.003 | summer | −0.340 | 0.004 | vict.black | −0.620 | 0.001 |
| winter | −0.249 | 0.019 | celebration | −0.639 | 0.007 | vict.black | 0.569 | 0.006 | summer | 0.304 | 0.005 |
| | | | lan.african | 0.119 | 0.027 | vict.female | 1.045 | 0.007 | vict.female | −0.940 | 0.006 |
| | | | | | | christian | −0.363 | 0.009 | pop.20.64 | −0.028 | 0.015 |
| | | | | | | pop.20.64 | 0.027 | 0.032 | vict.white | −0.377 | 0.030 |
| | | | | | | celebration | −0.527 | 0.035 | christian | 0.288 | 0.031 |
| | | | | | | lan.arabic | −0.105 | 0.042 | | | |
| | | | | | | lan.vietnamese | 0.098 | 0.044 | | | |
| | | | | | | poverty | −0.014 | 0.045 | | | |

**Table 14.** Model coefficients and p-values of significant ($p < 0.05$) predictors taken from linear models using $d_i^{\text{(day,week,month)}}$ and $\log(\Delta t_i)$ as dependent variables and using data from NY.

| NY | | | | | | | | | | | |
|---|---|---|---|---|---|---|---|---|---|---|---|
| dep. var. | $d_i^{\text{day}}$ | | | $d_i^{\text{week}}$ | | | $d_i^{\text{month}}$ | | | $\log(\Delta t_i)$ | |
| pred. var. | coeff. | p-val. | pred. var. | coeff. | p-val. | pred. var. | coeff. | p-val. | pred. var. | coeff. | p-val. |
| vict.u18 | −1.149 | 0.000 | federal | −0.859 | 0.000 | federal | −1.034 | 0.000 | federal | 1.156 | 0.000 |
| federal | −0.722 | 0.000 | vict.u18 | −1.093 | 0.000 | vict.u18 | −1.012 | 0.000 | vict.u18 | 1.444 | 0.000 |
| winter | −0.250 | 0.002 | winter | −0.260 | 0.001 | spring | 0.310 | 0.000 | winter | 0.496 | 0.000 |
| vict.black | 0.427 | 0.004 | vict.black | 0.440 | 0.002 | winter | −0.281 | 0.001 | vict.black | −0.748 | 0.000 |
| summer | 0.163 | 0.036 | vict.white | 0.382 | 0.019 | muslim | 0.930 | 0.001 | vict.white | −0.678 | 0.000 |
| lan.african | 0.082 | 0.043 | pop.no.insur | −0.079 | 0.026 | vict.black | 0.422 | 0.004 | muslim | −0.726 | 0.007 |
| vict.female | 0.615 | 0.049 | vict.25.44 | 0.400 | 0.032 | vict.white | 0.463 | 0.006 | spring | −0.237 | 0.017 |
| | | | pop.native | −0.156 | 0.036 | christian | −0.268 | 0.020 | vict.female | −0.777 | 0.020 |
| | | | spring | 0.168 | 0.044 | vict.25.44 | 0.425 | 0.031 | jewish | −0.388 | 0.021 |
| | | | | | | hindu | −0.442 | 0.037 | | | |

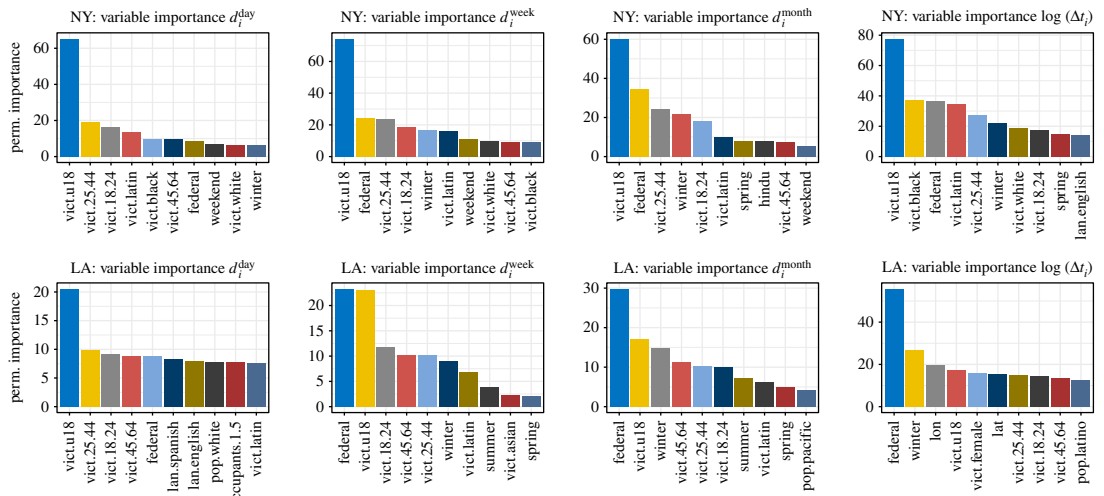

**Figure 5.** Predictor variable importance (permutation importance) of the 10 most important features for the different regression and classification tasks of the S3 model across both observed cities. Importance metrics are obtained from random forest models.

proportion of the population aged 20–64 corresponded to shorter delays. Some of the observed effects, for example, those related to survivor characteristics, confirm previous studies, while other findings from our expanded predictor pool, for example, the importance of holidays, are completely novel results.

Lastly, to confirm the findings of our coefficient analysis, we also assess the variable importance of each predictor, extracted from a global RF model. Here, we examine permutation importance, which is defined as the decrease in a model score (e.g. RMSE) when the variable of interest is randomly shuffled. Figure 5 highlights the 10 most important predictors for our models in both cities. We can see that these results are consistent with the coefficient analysis, highlighting the predictive value of the survivor's age, race and gender, as well as seasonality, holidays and local socio-economic characteristics.

# 4. Discussion and concluding remarks

In the present study, we propose a machine learning-based modelling framework to predict reporting delays of rape. While previous studies have mostly relied on surveys and other sensitive survivor data, our approach is based solely on openly available data. Beyond this, we add to the existing literature by expanding the set of previously used predictor variables, thus shedding new light on delay indicators. Our study confirms previous findings that survivor age and demographics relate to delayed reporting [16], and also contributes novel insights. We find that rape reporting delays exhibit spatial dependencies, which are not mitigated through adding survivor characteristics and temporal information on the occurrence date. This suggests that additional information might explain the spatial structure through an underlying social process. We confirm this hypothesis by deploying several implicit and explicit spatial models, accounting for (i) demographics and socio-economic factors, (ii) police district, and (iii) spatial coordinates. While the improvement in predictive power is relatively small, the spatial models successfully eliminate RSA in the model error terms. We further show that the combined model (S3), including socio-economic and demographic features as well as crime coordinates, consistently outperforms the non-spatial baselines. We also highlight novel insights into significant predictors of delayed reporting, confirming results from previous studies and presenting novel predictors such as crime occurrence on holidays. Overall, the additional predictor variables and spatial models expand substantially on previous studies that use sensitive survivor and event information [18].

As a result, we believe that our study can contribute to policy efforts that provide assistance to vulnerable communities. Our modelling approach aims to be comprehensive, transparent and focused on prediction. Hence, it is possible to implement our findings as part of public policy efforts tackling sexual violence and delayed reporting of rape. Since we can identify both vulnerable subpopulations and community areas, locally and temporally targeted interventions and long-term community programmes may help to prevent delayed reporting (also, most likely, reducing the risk of non-reporting) and hence support survivors of sexual violence. Beyond the public sector, NGOs focused

on preventing sexual violence and providing support to rape survivors can also benefit from our findings. All of the data used in this study is publicly available, which enables organizations without access to sensitive survivor information to deploy data-driven, informed programmes.

There are several avenues for further work following this research: our modelling approach is currently static, working with temporally aggregated data. Thus, we aim to examine the dynamics underlying the reporting of rape in the future, and in particular, the analysis of potential spatial contagion in reporting behaviour. If our findings regarding an underlying spatial process of rape reporting are confirmed, locally targeted policies could utilize the observed spillover effects to increase their reach.

Machine learning scholarship is currently subject to scrutiny in relation to transparent and fair predictions. Our dataset is an example of data that, although public, must be addressed with care and attention to detail. Behind every observation lies a human tragedy and unquantifiable suffering. When dealing with crime in general and sexual crimes in particular, exceptional attention should be paid to protecting survivors, in particular in avoiding survivor re-identification.

While we saw similar sets of significant predictors across the cities of New York and Los Angeles, our proposed models nevertheless have limited scope and ability to predict across cities. Fundamentally, the model's performance must be contextualized by the datasets included and the environments in which these data were gathered. For instance, the large cities of New York and Los Angeles differ in jurisdictions, policing practices and social and physical structures. Findings from our empirical analysis cannot be assumed to be general, particularly those in relation to sensitive demographic factors such as ethnicity and socio-economic status. While we find these to be important predictors, particularly in New York, further studies assessing these effects should be conducted and new predictors should be tested. Additionally, our dataset does not include attempted rapes or rape committed outside of a domestic setting, nor does it include unreported rape crimes or those that were reported outside the temporal range of our data. We encourage other researchers to use the same datasets to test new modelling approaches, as well as expanding the set of cities, the time range and the set of predictive variables considered in the analysis.

Data accessibility. The data used for this study is publicly available, via the Open Data portals of New York City and Los Angeles and the American Community Survey. These can be accessed as follow: New York crime data: https://data.cityofnewyork.us/Public-Safety/NYPD-Complaint-Data-Historic/qgea-i56i. Los Angeles crime data: https://data.lacity.org/A-Safe-City/Crime-Data-from-2010-to-Present/63jg-8b9z. American Community Survey: https://data.census.gov/cedsci/.

Authors' contributions. K.K., D.B.N. and S.A.J. conceptualized the paper. K.K. and D.B.N. developed the methodological framework. K.K. assembled the data, conducted the analysis and wrote the first manuscript. D.B.N. and S.A.J. validated the analysis and edited the paper.

Competing interests. There are no competing interests.

Funding. The authors gratefully acknowledge funding from the UK Engineering and Physical Sciences Research Council, the EPSRC Centre for Doctoral Training in Urban Science (EPSRC grant no. EP/L016400/1); the Alan Turing Institute (EPSRC grant no. EP/N510129/1).

Acknowledgements. We want to thank Calvin Li for his valuable support for gathering and filtering the public crime records. We want to thank the participants of the 2018 NeurIPS workshop on 'Decision making in the spatio-temporal domain', where a preliminary work-in-progress version of this study was presented, for their valuable feedback. This preliminary version can be accessed via arXiv (see here: https://arxiv.org/pdf/1905.09796.pdf)

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
