## [Peer Review File · Royal Society Open Science]

Review History

RSOS-200156.R0 (Original submission)

Review form: Reviewer 1

Is the manuscript scientifically sound in its present form?

No

Are the interpretations and conclusions justified by the results?

Yes

Is the language acceptable?

Yes

Do you have any ethical concerns with this paper?

No

Have you any concerns about statistical analyses in this paper?

Yes

Recommendation?

Major revision is needed (please make suggestions in comments)

Comments to the Author(s)

This paper aims at uncovering patterns in delayed rape reporting. In order to achieve this, data from two major U.S. cities are utilized within a machine learning framework. The topic is of high importance, and is thus of good fit to RSOS. However, the overall analysis fairly trivial and does not follow a clear line of thought, it simply looks like somebody has combined different python functions from a tutorial without a clear objective.

While this research covers a rather novel and under-researched topic, there are multiple weak points that need to be addressed in a revision before resubmission.

- A clear research objective is missing. On the one hand, the paper aims at making accurate predictions. Why is this needed? How can this help to improve policies? On the other hand, a deep understanding of underlying (causal) factors is sought to improve governmental policy. How are variables contributing to the overall patterns (i.e. variable importance does not inform me on which variables to act; rather, I would search for effect sizes etc.)? Neither of those question is answered sufficiently. My recommendation is to define a clear research goal (i.e. understanding OR prediction) and perform the analysis more extensively.

* The paper claims that the work has implications, yet they remain fairly vague. I strongly recommend to elaborate them more extensively. As the black-box nature is not resolved, I don't think that -- on the understanding part -- a policy maker would know which variables should be targeted by policies (e.g. a counterfactual analysis could be a way forwardd). This could be a line of thought that could be improved. If a prediction approach is followed, the authors would need to argue why the prediction is necessary in the first place (e.g. some states do not collect rape delays, so the mechanism could be transferred to a different state; this would require that the authors test the between-city convertability of their prediction). Again, I would recommend to choose one path and describe that more clearly.

* I personally find the spatial autocorrelation somewhat unneeded and it can thus be removed. What is more important is that the findings of the spatial autocorrelation are not taken properly into consideration during modeling. That is, their machine learning model does not follow state-of-the-art models for spatial heterogeneity (eg here: <https://www.nature.com/articles/s41586-019-1878-8>). I would also discuss the size of the spatial residuals as they inform to what extent reporting variables cannot capture the full variance in y.

* If "understanding" is followed: Would it make sense to model the heavy-tailed distribution of y via a suitable hierarchical GLM directly? Generally, the paper could be improved by providing some theoretical underpinnings and justification for modeling choices rather than just stating "extensive experiments".

- The results often remain shallow and the model performance is only tested against an uninformed guess instead of a proper baseline. If prediction is followed, I would group variables into different bins (e.g. socio-economic, spatial, etc.) and then compare how the prediction accuracy changes (Cohen's f^2).

* Why are variable importance relevant at all?

- The article should point out its limitations more clearly. This includes the repeated use of strong assumptions as well as limited generalizability and interpretability of the model.

Minor

* A clearer presentation (e.g. variables as tables) could also make the paper easier to follow.

* Currently, the structure is a bit odd (model interpretation via variable importance factors before reporting the model itself; descriptives quite isolated from the data description; spatial analysis spread throughout the paper instead of being bundled).

* The authors state the the factors in fig 3 "explain" .. I would argue that correlation plats without controlling for confounders are the opposite of explaining.

* Why need for two aggregated + disaggregated models?

* Please check the numbering of figures / tables.

Review form: Reviewer 2

Is the manuscript scientifically sound in its present form?

No

Are the interpretations and conclusions justified by the results?

No

Is the language acceptable?

Yes

Do you have any ethical concerns with this paper?

No

Have you any concerns about statistical analyses in this paper?

Yes

Recommendation?

Reject

Comments to the Author(s)

See attachment (Appendix A).

Decision letter (RSOS-200156.R0)

27-Feb-2020

Dear Mr Klemmer:

Manuscript ID RSOS-200156 entitled "Understanding Spatial Patterns in Rape Reporting Delays" which you submitted to Royal Society Open Science, has been reviewed. The comments from reviewers are included at the bottom of this letter.

In view of the criticisms of the reviewers, the manuscript has been rejected in its current form. However, a new manuscript may be submitted which takes into consideration these comments.

Please note that resubmitting your manuscript does not guarantee eventual acceptance, and that your resubmission will be subject to peer review before a decision is made.

Your resubmitted manuscript should be submitted by 26-Aug-2020. If you are unable to submit by this date please contact the Editorial Office.

on behalf of Dr Cecilia Mascolo (Associate Editor) and Marta Kwiatkowska (Subject Editor)
openscience@royalsociety.org

Associate Editor Comments to Author (Dr Cecilia Mascolo):

Comments to the Author:

The reviewers consistently find the paper topic interesting however they raise a number of concerns with respect to the methodology, the evaluation framework and the presentation. As a consequence, the paper cannot be accepted in its current form.

Reviewers' Comments to Author:

Reviewer: 1

Comments to the Author(s)

This paper aims at uncovering patterns in delayed rape reporting. In order to achieve this, data from two major U.S. cities are utilized within a machine learning framework. The topic is of high importance, and is thus of good fit to RSOS. However, the overall analysis fairly trivial and does not follow a clear line of thought, it simply looks like somebody has combined different python functions from a tutorial without a clear objective.

While this research covers a rather novel and under-researched topic, there are multiple weak points that need to be addressed in a revision before resubmission.

- A clear research objective is missing. On the one hand, the paper aims at making accurate predictions. Why is this needed? How can this help to improve policies? On the other hand, a deep understanding of underlying (causal) factors is sought to improve governmental policy. How are variables contributing to the overall patterns (i.e. variable importance does not inform me on which variables to act; rather, I would search for effect sizes etc.)? Neither of those question is answered sufficiently. My recommendation is to define a clear research goal (i.e. understanding OR prediction) and perform the analysis more extensively.

* The paper claims that the work has implications, yet they remain fairly vague. I strongly recommend to elaborate them more extensively. As the black-box nature is not resolved, I don't think that -- on the understanding part -- a policy maker would know which variables should be targeted by policies (e.g. a counterfactual analysis could be a way forward). This could be a line of thought that could be improved. If a prediction approach is followed, the authors would need to argue why the prediction is necessary in the first place (e.g. some states do not collect rape delays, so the mechanism could be transferred to a different state; this would require that the authors test the between-city convertibility of their prediction). Again, I would recommend to choose one path and describe that more clearly.

* I personally find the spatial autocorrelation somewhat unneeded and it can thus be removed. What is more important is that the findings of the spatial autocorrelation are not taken properly

into consideration during modeling. That is, their machine learning model does not follow state-of-the-art models for spatial heterogeneity (eg here: <https://www.nature.com/articles/s41586-019-1878-8>). I would also discuss the size of the spatial residuals as they inform to what extent reporting variables cannot capture the full variance in y .

* If "understanding" is followed: Would it make sense to model the heavy-tailed distribution of y via a suitable hierarchical GLM directly? Generally, the paper could be improved by providing some theoretical underpinnings and justification for modeling choices rather than just stating "extensive experiments".

- The results often remain shallow and the model performance is only tested against an uninformed guess instead of a proper baseline. If prediction is followed, I would group variables into different bins (e.g. socio-economic, spatial, etc.) and then compare how the prediction accuracy changes (Cohen's f_2).

* Why are variable importance relevant at all?

- The article should point out its limitations more clearly. This includes the repeated use of strong assumptions as well as limited generalizability and interpretability of the model.

Minor

* A clearer presentation (e.g. variables as tables) could also make the paper easier to follow.

* Currently, the structure is a bit odd (model interpretation via variable importance factors before reporting the model itself; descriptives quite isolated from the data description; spatial analysis spread throughout the paper instead of being bundled).

* The authors state the the factors in fig 3 "explain" .. I would argue that correlation plots without controlling for confounders are the opposite of explaining.

* Why need for two aggregated + disaggregated models?

* Please check the numbering of figures / tables.

Reviewer: 2

Comments to the Author(s)

See attachment ("RoyalSocietu.pdf")

Author's Response to Decision Letter for (RSOS-200156.R0)

See Appendix B.

RSOS-201795.R0

Review form: Reviewer 1

Is the manuscript scientifically sound in its present form?

Yes

Are the interpretations and conclusions justified by the results?

Yes

Is the language acceptable?

Yes

Do you have any ethical concerns with this paper?

No

Have you any concerns about statistical analyses in this paper?

No

Recommendation?

Accept with minor revision (please list in comments)

Comments to the Author(s)

The paper has overall greatly improved, now coming up with a clear, consistent and actionable story line.

Besides that, I would suggest the following changes for a better presentation:

- 1) The "background" materials on Moran's I are a bit odd. An informed reader should now them and, if not, why are not all other models like a GP explained step-by-step? Long story short, I would suggest to remove it or, if the author(s) prefer, move it to the appendix.
- 2) Sections 2.1-2-3 are summary/descriptive statistics, not results from their models. Hence, I would suggest to split this section into two, one for "Summary statistics" and one for "Results".
- 3) "Explicit spatial model": I wonder if this is an appropriate term? Reading that name, I would have expected the FE model to be called "explicit spatial model". Maybe a term like "Covariate" or "Census spatial model" are more of fit?
- 4) After carefully reading the manuscript, I found the overall description of the outcome variables highly inconsistent. I recommend that the authors take a thorough look and streamline this. For instance, they extensively discuss why a thresholded d is important. However, later, they work on $\log(t)$, which hit me by surprise. At the same time, they frequently jump back and forth between p and d .

For a better comparison, I would suggest adding a naive baseline (majority vote) to the results tables. I assume that their outcome variable is skewed and, hence, the accuracy must be treated with caution. Here the naive baseline could facilitate a better interpretation of the results.

Given the new positioning of the work, I can live with the feature importance. However, I am concerned with the regression results (which estimator? OLS? multicollinearity? could the latter conflate p-values? no discussion on this?). Even more important, a journal on data science like RSOS should not advocate frequentist statistics given the obvious shortcomings; instead, I would suggest a Bayesian estimator (see Gelman's BDA3 and the surrounding discussion). That being said, I would be fine with different roads that the authors could take (e.g., even removing this estimation, given that their story is on the overall predictive utility of different variable groups and, against this background, the coefficient estimates are not necessary; If they want the coeffs to be included, I would advocate a plot).

Overall, the discussion could benefit from a more truthful statement on the results: the prediction performance changes only barely across the models. Again, this is not "bad" but a finding (a relevant one) of its own. Maybe by reflecting upon the impact (e.g. how many cases would be correctly predicted additionally when changing from model NS-x to S-y) could put the results more into perspective.

Review form: Reviewer 3

Is the manuscript scientifically sound in its present form?

Yes

Are the interpretations and conclusions justified by the results?

Yes

Is the language acceptable?

Yes

Do you have any ethical concerns with this paper?

No

Have you any concerns about statistical analyses in this paper?

No

Recommendation?

Accept with minor revision (please list in comments)

Comments to the Author(s)

The manuscript makes a clear contribution to the literature on predicting reporting delays, especially delays in rape reporting. Given its framing, which emphasizes the literature's previous inattention to spatial and temporal factors, it seems more could be said about ways in which human activities structure time and space to generate the conditions under which the forecasting model picks up these relationships. If the authors wish to put in more of this framing material, both in the introduction and the discussion, they might usefully examine the sociological literature on neighborhood effects. Likewise, the criminological literature on routine activities theory and situational crime prevention could offer some additional interpretations for both the spatial and temporal results noted.

Decision letter (RSOS-201795.R0)

Dear Mr Klemmer

On behalf of the Editors, we are pleased to inform you that your Manuscript RSOS-201795 "Understanding Spatial Patterns in Rape Reporting Delays" has been accepted for publication in Royal Society Open Science subject to minor revision in accordance with the referees' reports. Please find the referees' comments along with any feedback from the Editors below my signature.

Please submit your revised manuscript and required files (see below) no later than 7 days from today's (ie 10-Dec-2020) date. Note: the ScholarOne system will 'lock' if submission of the revision is attempted 7 or more days after the deadline. If you do not think you will be able to meet this deadline please contact the editorial office immediately.

Best regards,

on behalf of Professor Marta Kwiatkowska (Subject Editor)
openscience@royalsociety.org

Reviewer comments to Author:

Reviewer: 1

Comments to the Author(s)

The paper has overall greatly improved, now coming up with a clear, consistent and actionable story line.

Besides that, I would suggest the following changes for a better presentation:

- 1) The "background" materials on Moran's I are a bit odd. An informed reader should now them and, if not, why are not all other models like a GP explained step-by-step? Long story short, I would suggest to remove it or, if the author(s) prefer, move it to the appendix.
- 2) Sections 2.1-2-3 are summary/descriptive statistics, not results from their models. Hence, I would suggest to split this section into two, one for "Summary statistics" and one for "Results".
- 3) "Explicit spatial model": I wonder if this is an appropriate term? Reading that name, I would have expected the FE model to be called "explicit spatial model". Maybe a term like "Covariate" or "Census spatial model" are more of fit?
- 4) After carefully reading the manuscript, I found the overall description of the outcome variables highly inconsistent. I recommend that the authors take a thorough look and streamline this. For instance, they extensively discuss why a thresholded d is important. However, later, they work on $\log(t)$, which hit me by surprise. At the same time, they frequently jump back and forth between p and d .

For a better comparison, I would suggest adding a naive baseline (majority vote) to the results tables. I assume that their outcome variable is skewed and, hence, the accuracy must be treated with caution. Here the naive baseline could facilitate a better interpretation of the results.

Given the new positioning of the work, I can live with the feature importance. However, I am concerned with the regression results (which estimator? OLS? multicollinearity? could the latter

conflate p-values? no discussion on this?). Even more important, a journal on data science like RSOS should not advocate frequentist statistics given the obvious shortcomings; instead, I would suggest a Bayesian estimator (see Gelman's BDA3 and the surrounding discussion). That being said, I would be fine with different roads that the authors could take (e.g., even removing this estimation, given that their story is on the overall predictive utility of different variable groups and, against this background, the coefficient estimates are not necessary; If they want the coefs to be included, I would advocate a plot).

Overall, the discussion could benefit from a more truthful statement on the results: the prediction performance changes only barely across the models. Again, this is not "bad" but a finding (a relevant one) of its own. Maybe by reflecting upon the impact (e.g. how many cases would be correctly predicted additionally when changing from model NS-x to S-y) could put the results more into perspective.

Reviewer: 3

Comments to the Author(s)

The manuscript makes a clear contribution to the literature on predicting reporting delays, especially delays in rape reporting. Given it's framing, which emphasizes the literature's previous inattention to spatial and temporal factors, it seems more could be said about ways in which human activities structure time and space to generate the conditions under which the forecasting model picks up these relationships. If the authors wish to put in more of this framing material, both in the introduction and the discussion, they might usefully examine the sociological literature on neighborhood effects. Likewise, the criminological literature on routine activities theory and situational crime prevention could offer some additional interpretations for both the spatial and temporal results noted.

===PREPARING YOUR MANUSCRIPT===

If you have been asked to revise the written English in your submission as a condition of publication, you must do so, and you are expected to provide evidence that you have received language editing support. The journal would prefer that you use a professional language editing service and provide a certificate of editing, but a signed letter from a colleague who is a native

speaker of English is acceptable. Note the journal has arranged a number of discounts for authors using professional language editing services (<https://royalsociety.org/journals/authors/benefits/language-editing/>).

===PREPARING YOUR REVISION IN SCHOLARONE===

-- If you have uploaded ESM files, please ensure you follow the guidance at <https://royalsociety.org/journals/authors/author-guidelines/#supplementary-material> to include a suitable title and informative caption. An example of appropriate titling and captioning

may be found at https://figshare.com/articles/Table_S2_from_Is_there_a_trade-off_between_peak_performance_and_performance_breadth_across_temperatures_for_aerobic_sc_ope_in_teleost_fishes_/3843624.

Author's Response to Decision Letter for (RSOS-201795.R0)

See Appendix C.

RSOS-201795.R1 (Revision)

Review form: Reviewer 1

Is the manuscript scientifically sound in its present form?

Yes

Are the interpretations and conclusions justified by the results?

Yes

Is the language acceptable?

Yes

Do you have any ethical concerns with this paper?

No

Have you any concerns about statistical analyses in this paper?

No

Recommendation?

Accept as is

Comments to the Author(s)

Congrats, I can recommend acceptance.

Decision letter (RSOS-201795.R1)

Dear Mr Klemmer,

It is a pleasure to accept your manuscript entitled "Understanding Spatial Patterns in Rape Reporting Delays" in its current form for publication in Royal Society Open Science. The comments of the reviewer(s) who reviewed your manuscript are included at the foot of this letter.

on behalf of Prof Marta Kwiatkowska (Subject Editor)
openscience@royalsociety.org

Reviewer comments to Author:
Reviewer: 1

Comments to the Author(s)
Congrats, I can recommend acceptance.

Appendix A

Review of “Understanding Spatial Patterns of Rape Reporting Delays”

February 22, 2020

This paper tackles an important issue with interesting data. The writing largely is clear and efficient. I also like the figures. However, I don't think the authors pull it off. Here are the problems I see.

1. The case for defining a categorical outcome variable (page 4) is not formally justified except as a matter of convenience. There is no statistical problem with a heavy tailed distributions as long as they are an accurate representation of the reality. But if the authors find this troubling nevertheless, a log or square root transformation should help a lot. By choosing the make a key part of analysis a classification exercise, the analysis is just more involved unnecessarily.
2. As the authors note, most rapes are not reported at all. That raises a question of how the reported rapes are different and, therefore, to what population we are to make inferences. There is also censoring for cases that had such long reporting delays that they are no in the dataset. This is a classic problem in biomedical research that needs to at least be acknowledged. Unfortunately, the authors do not have the data to address its impact.
3. The p-values used for Moran's I make no sense to me (page 6). The data are a population or perhaps a convenience sample. There is no proposed model – either a statistical model or a substantive model – of the data generation process. So, either there is no uncertainty to address or its nature is unknown. In either case, those p-values have no proper statistical interpretation. They are just a ritual. The later permutation tests (page 9) underscore this point because they must assume a

fixed set of observational units to which all inference are limited. If the permutation distribution makes for a sensible null hypothesis (justification needs to be provided), those p-values, in contrast, have a clear interpretation.

4. The two-step approach on page 7 is ill advised. The impact of socio-economic and demographic factors is necessarily overestimated because they are correlated with the spatial context. The “overlap” gets allocated to the socio-economic and demographic factors. In effect, the authors are partitioning the explained variation in a way which disadvantages spatial context. For what it is worth, this is an old problem on which there is a large literature in the social sciences. Bottom line: there are many ways to partition the variance that give you different conclusions. The problem stems from the covariances that cannot be partitioned.
5. Around page 9, the authors seem confused or the writing is unclear. Algorithms are *not* models although I appreciate that that term is often used (incorrectly). Because they are not models, it is very risky to make claims about which predictors matter and how much. In this instance, moreover, the authors use as a measure of importance the so-called “Gini importance metric,” which is just an average of the contribution to the fit over the tree ensemble. Why is contribution to fit a measure of importance unless one proceeds with a tautology? It is *not* a measure, for example, of any causal impact on the actual outcome class (despite how the authors write) or on forecasting accuracy. It also averages over all outcome classes which is misleading because the role of predictors will vary depending on the outcome class. And for random forests at least, one can very easily obtain the impact of each predictor on forecasting accuracy separately for each outcome class. Why not use that? In this way, one acknowledges that a random forest is not model and yet we can use it properly for forecasting. We also need more detail on the gradient boosting used and on the assumptions made for the gaussian process models. For the latter, there are many reasonable ways to smooth spatial data. Why this? And what is being assumed?
6. It seems to me that the authors have a conventional hierarchical data structure. Rape cases as nested within spatial units. Why not proceed with a hierarchical statistical formulation? There is a large literature

here too in spatial statistics and econometrics. If explanation is the goal a (real) model of some sort is required.

In summary, it looks like the authors fail to distinguish between explanation and prediction. They try to do both at the same time with algorithms. Algorithms can be great for prediction. You need models (or experiments) for explanation. Starting with the very title of the paper, the authors cannot deliver. I should add that if the authors take a close look at the best work on predictive policing, the goal is prediction not explanation. For example, they cite Flaxman and Loeffler favorably. They will find no explanation in that work. If they authors really want to explain the factors that affect reporting delays, they will need a model. I urge the authors to take on that task.

Appendix B

Dear RSOS Editors and Reviewers,

Thank you for your extensive comments on our submission and for the opportunity to resubmit our work to RSOS. According to the reviewers' feedback, we have substantially revised our manuscript to address these concerns. The predictive modelling section has been rewritten to focus only on the disaggregated (case-level) data, considering the non-spatial variables (survivor¹ characteristics and temporal features) in our baseline models (NS1-NS2) for prediction, and examining both the performance gains and the reduction in residual spatial autocorrelation (RSA) as spatial information is incorporated in various ways (spatial models S1-S3). We clearly delineate between the five models (each assuming the dependence of rape reporting delays on a different set of predictor variables) and the four statistical and machine learning methods (algorithms) used to fit these models (L1-penalized linear regression; random forests; gradient-boosted trees; and Gaussian processes). This revised paper structure, comparing non-spatial models against implicit and explicit spatial models, allows us to gain understanding of the observed spatial dependencies in reporting delays. We find that predicting reporting delays using survivor characteristics and temporal features achieves reasonably good prediction performance, but leaves spatially autocorrelated residuals. Incorporating spatial information, either by controlling for the socio-economic and demographic features of the local area, or by modelling the spatial correlation explicitly with Gaussian processes, eliminates the RSA and improves predictive performance, with the best performance being achieved by the combined model (S3). Moreover, we examine the (multivariate) linear model coefficients and identify the significant predictors, thus both predicting rape reporting delays and contributing to the understanding of which variables are significantly associated with delays.

We do want to clarify that the main objective of the paper remains prediction, not causal modelling (there is not an identification strategy, and correlation/association does not imply causation). We have adapted our manuscript to make it clear that our experiments are solely focused on predictive modelling and examining associations between delays in rape reporting and spatial, temporal, demographic, and socio-economic features. We have further expanded our reasoning as to how this predictive modelling approach may influence policy design: the novel associations shown in this study provide a basis for providing targeted services at the right place and at the right time. Identifying vulnerable subpopulations (e.g. people living in crowded housing situations) and high-risk periods (e.g. winter holiday season) may not only help to make interventions more effective, but might pave the way for more comprehensive modelling of rape reporting delays.

We now provide a point by point response to the remaining issues raised by the reviewers (highlighted in blue), with our responses to each point in italics.

Reviewer 1:

A clear research objective is missing. On the one hand, the paper aims at making accurate predictions. Why is this needed? How can this help to improve policies? On the other hand, a deep understanding of underlying (causal) factors is sought to improve governmental policy. How are variables contributing to the overall patterns (i.e. variable importance does not inform me on which variables to act; rather, I would search for effect sizes etc.)? Neither of those question is answered

¹Please note that throughout the revised text we now refer to persons who reported rape crimes committed against their person as "survivors" rather than "victims". Based on the literature, we believe this term better reflects the preferred self-identification of rape survivors.

sufficiently. My recommendation is to define a clear research goal (i.e. understanding OR prediction) and perform the analysis more extensively.

We have substantially revised the methods and results sections to address this concern. We are not making strong causal claims: we are identifying associations only.

The goals of the paper are three-fold: (1) to make more accurate predictions through building better predictive models, including choice of predictor variables and statistical/machine learning methods; (2) to understand which subsets of predictor variables are or are not sufficient to explain the spatial variation in rape reporting delays; and (3) to determine which variables are significant predictors of reporting delays.

Goal (1) is achieved through a comparison of the predictive accuracy of five models (two non-spatial and three spatial), each with a different set of predictor variables, using four different statistical and machine learning methods (algorithms) to fit these models. The combined models incorporating spatial information achieve high prediction accuracy and provide a strong basis for targeting interventions to different demographics and geographic areas.

Goal (2) is achieved by examining the residual spatial autocorrelation from non-spatial and spatial models; we observe that controlling for survivor characteristics (with or without temporal features) is not sufficient to remove the spatial autocorrelation, but additionally controlling for demographic and socioeconomic features of the local area is sufficient. Thus we are able to explain the spatial variation in reporting delays through the spatial variation in the demographics of rape survivors and the socio-economic and demographic characteristics of their surrounding neighbourhoods.

Goal (3) is achieved through identification of significant predictor variables, and their model coefficients, in our L1-penalized linear models. We observe that survivor characteristics (age under 18, non-female, Asian or Latino as compared to White or Black survivors), temporal characteristics (winter season; holidays), and socio-economic characteristics of the local area (poverty; uninsured and underinsured populations) are predictive of longer reporting delays, thus allowing us to target specific demographics and assist vulnerable subpopulations.

The results often remain shallow and the model performance is only tested against an uninformed guess instead of a proper baseline. If prediction is followed, I would group variables into different bins (e.g. socio-economic, spatial, etc.) and then compare how the prediction accuracy changes (Cohen's f^2).

This is an excellent idea, thanks so much for suggesting! We now present five different models, each with different groups of predictor variables. Non-spatial model NS1 includes only survivor characteristics; non-spatial model NS2 includes NS1 + temporal features; spatial model S1 includes NS2 + socio-economic and demographic features of the local area; spatial model S2 includes NS2 + police precinct dummies; and spatial model S3 includes S1 + explicit spatial coordinates, using Gaussian processes to model spatial correlation.

I personally find the spatial autocorrelation somewhat unneeded and it can thus be removed. What is more important is that the findings of the spatial autocorrelation are not taken properly into consideration during modeling. That is, their machine learning model does not follow state-of-the-art models for spatial heterogeneity (eg here: <https://www.nature.com/articles/s41586-019-1878-8>). I would also discuss the size of the spatial residuals as they inform to what extent reporting variables cannot capture the full variance in y .

The Reviewer argues that there is a lack of need for analysis of spatial autocorrelation in the context of our paper. We disagree with the reviewer in this regard, as we believe that our original analysis of spatial dependencies in reporting delays is particularly interesting: it implies unaccounted for predictor variables that the remainder of our paper attempts to explain through predictive modelling. This phenomenon has previously not been studied with rape reporting delays. In the revised paper, we further show that incorporating survivor characteristics and temporal features is not sufficient to eliminate spatial dependencies; moreover, adding the spatial features not only eliminates residual spatial autocorrelation, but also improves predictive performance.

If "understanding" is followed: Would it make sense to model the heavy-tailed distribution of y via a suitable hierarchical GLM directly? Generally, the paper could be improved by providing some theoretical underpinnings and justification for modeling choices rather than just stating "extensive experiments".

In our revised paper, we address this concern in several ways: (1) we incorporate hierarchy by controlling for the socio-economic and demographic features of the local area in spatial model S1; including dummies for the local area (police precinct) in spatial model S2; and including explicit spatial coordinates (and modelling spatial correlation using Gaussian processes) in spatial model S3; (2) we focus more on linear models (L1-penalized linear and logistic regression) in addition to the machine learning models, thus allowing us to assess significance of model coefficients; and (3) to account for the heavy-tailed nature of the reporting delays, we include experiments using log-transformed reporting delays as well as the categorical outcome variables we originally considered.

Why are variable importance relevant at all?

We believe that our examination of variable importance provides two important contributions: (1) We find novel, previously unknown associations between rape reporting delays and several temporal and socio-economic features, e.g. household density or holiday season crime occurrence. (2) The found associations can be used as a basis for policy design and service provision, as well as motivate further methodological studies into these processes. The variable importance analysis is expanded substantially from the original manuscript and now focuses primarily on an examination of significant linear model coefficients, with random forest variable importance de-emphasized but shown to support and reinforce the linear model results.

Reviewer 2:

The case for defining a categorical outcome variable (page 4) is not formally justified except as a matter of convenience. There is no statistical problem with a heavy tailed distributions as long as they are an accurate representation of the reality. But if the authors find this troubling nevertheless, a log or square root transformation should help a lot. By choosing the make a key part of analysis a classification exercise, the analysis is just more involved unnecessarily.

The revised manuscript expands on the justification for using delay thresholds of one day, one week and one month respectively: the decision of using these thresholds does not only stem from their heuristic value, but is also motivated by insights from forensics and mental health research. Survivors reporting and seeking support within one day have been shown to have better treatment outcomes [1], while their medical examination can provide the best possible forensic evidence [2] to assist persecution. For female survivors of sexual assault with penetration, traces of DNA evidence last for about one week due to vaginal drainage, menstruation and sperm degradation [2]. Lastly, rape reported one month or longer after the occurrence is associated with a severe increase in psychological distress for survivors [3]. One month is also the approximate period to allow for the

recovery of forensic evidence of potential drugging from the survivors hair [4]. The reviewers also suggest the use of log-transformation as an option to deal with the heavily tailed reporting delay data. We have expanded our manuscript accordingly to include experiments with log-transformed reporting delays.

As the authors note, most rapes are not reported at all. That raises a question of how the reported rapes are different and, therefore, to what population we are to make inferences. There is also censoring for cases that had such long reporting delays that they are not in the dataset. This is a classic problem in biomedical research that needs to at least be acknowledged. Unfortunately, the authors do not have the data to address its impact.

While we agree that we do not have data that would allow us to fully assess the degree of non-reporting (or reports that are delayed past the endpoint of the data), we rely on the prior findings that under-reporting and delayed reporting are highly correlated, and thus including the "missing" cases would not be expected to substantially change the relationship between predictors and target variables. Additionally, we do not set an "earliest" occurrence date so we account for all rapes in the data, even those with many decades long reporting delays which are eventually reported in the chosen time frame. Our sub-selection orients itself solely by the reporting date.

The p-values used for Moran's I make no sense to me (page 6). The data are a population or perhaps a convenience sample. There is no proposed model – either a statistical model or a substantive model – of the data generation process. So, either there is no uncertainty to address or its nature is unknown. In either case, those p-values have no proper statistical interpretation. They are just a ritual. The later permutation tests (page 9) underscore this point because they must assume a fixed set of observational units to which all inferences are limited. If the permutation distribution makes for a sensible null hypothesis (justification needs to be provided), those p-values, in contrast, have a clear interpretation.

The Moran's I p-values are based on a permutation test, where the observed neighbourhood configuration is tested against randomly assigned neighbourhoods. Moran's I and its p-values are a well-established method in spatial statistics and have proven to identify both spatial outliers and homogeneous spatial clusters. In the revised manuscript we have included more extensive results from spatial autocorrelation testing, providing strong evidence of the observed spatial dependencies both in the outcome variable (reporting delays) and model residuals (for non-spatial models NS1 and NS2). Further, we note that the Moran's I and the spatial permutation test on page 9 complement each other by testing different null hypotheses: the Moran's I tests for correlation between neighbouring cases, where the null hypothesis is complete spatial randomness (or CSR of the residuals when looking at residual spatial autocorrelation), while the spatial permutation test looks for deviations in the distribution of median reporting times among precincts, where the null hypothesis is random assignment of cases to precincts.

The two-step approach on page 7 is ill advised. The impact of socio-economic and demographic factors is necessarily overestimated because they are correlated with the spatial context. The "overlap" gets allocated to the socio-economic and demographic factors. In effect, the authors are partitioning the explained variation in a way which disadvantages spatial context. For what it is worth, this is an old problem which there is a large literature in the social sciences. Bottom line: there are many ways to partition the variance that give you. The problem stems from the covariances that cannot be partitioned.

We have eliminated the two-step approach from the revised paper. Now our only Gaussian process model (S3) incorporates both the socio-economic and demographic factors of the local area as well as explicitly modelling spatial correlation using the case-level spatial coordinates, and we demonstrate that this combined model often improves predictive accuracy.

Around page 9, the authors seem confused or the writing is unclear. Algorithms are not models although I appreciate that that term is often used (incorrectly). Because they are not models, it is very risky to make claims about which predictors matter and how much. In this instance, moreover, the authors use as a measure of importance the so-called "Gini importance metric," which is just an average of the contribution to the fit over the tree ensemble. Why is contribution to fit a measure of importance unless one proceeds with a tautology? It is not a measure, for example, of any causal impact on the actual outcome class (despite how the authors write) or on forecasting accuracy. It also averages over all outcome classes which is misleading because the role of predictors will vary depending on the outcome class. And for random forests at least, one can very easily obtain the impact of each predictor on forecasting accuracy separately for each outcome class. Why not use that? In this way, one acknowledges that a random forest is not model and yet we can use it properly for forecasting. We also need more detail on the gradient boosting used and on the assumptions made for the gaussian process models. For the latter, there are many reasonable ways to smooth spatial data. Why this? And what is being assumed?

In the revised manuscript, we have addressed this concern in several ways: (1) we instead focus on the significant predictors and their model coefficients from the linear models, with variable importance calculations from the random forest de-emphasized (and included only to confirm and support the findings from the linear models); (2) we replace the Gini importance with permutation importance, which is a measure of change in forecasting accuracy when that variable is permuted; (3) we carefully distinguish between the five models (each with a different subset of predictor variables) and the four algorithms used to fit models to data (penalized linear and logistic regression; random forests; gradient boosting; Gaussian processes), showing results for each combination of model and algorithm; (4) we make it clear that we see variable importance as an analysis of association and explicitly not as an indication of a causal relationship.

References

1. Miller, T. R., Cohen, M. A., & Wiersema, B. (1996). Survivor costs and consequences: A new look (NCJ 155282). Retrieved from the U.S. Department of Justice, Office of Justice Programs, National Institute of Justice: <https://www.ncjrs.gov/pdffiles/victcost.pdf>.
2. Cohen, M. A., & Piquero, A. R. (2009). New evidence on the monetary value of saving a high risk youth. *Journal of Quantitative Criminology*, 25(1), 25-49.
3. Cohen, M. A., Rust, R. T., Steen, S., & Tidd, S. T. (2004). Willingness-to-pay for crime control programs. *Criminology*, 42(1), 89-110.
4. McCollister, K. E., French, M. T., & Fang, H. (2010). The cost of crime to society: New crime-specific estimates for policy and program evaluation. *Drug and alcohol dependence*, 108(1-2), 98-109.

Appendix C

Thanks again to the associate editor and to the reviewers for their helpful comments, which have greatly improved the quality of the paper. Below please find a point-by-point response to the remaining minor issues raised by the reviewers.

Reviewer: 1

1) The "background" materials on Moran's I are a bit odd. An informed reader should know them and, if not, why are not all other models like a GP explained step-by-step? Long story short, I would suggest to remove it or, if the author(s) prefer, move it to the appendix.

We agree with the reviewer that the background discussion on Moran's I in its current state is too lengthy, and have shortened this paragraph accordingly.

2) Sections 2.1-2-3 are summary/descriptive statistics, not results from their models. Hence, I would suggest to split this section into two, one for "Summary statistics" and one for "Results".

We agree with the reviewer and have adapted our headings as follows:

2. Results

2.1 Summary statistics

2.1.1 Examining reporting delays

2.1.2 Spatial patterns in reporting behavior

2.1.3 Temporal patterns in reporting behavior

2.2 Forecasting reporting delays

2.2.1 Predictive performance

2.2.2 Assessing feature importance

3) "Explicit spatial model": I wonder if this is an appropriate term? Reading that name, I would have expected the FE model to be called "explicit spatial model". Maybe a term like "Covariate" or "Census spatial model" are more of fit?

We agree that the term "explicit spatial model" might lead to confusion and have changed the name of model S3 to "combined spatial model".

4) After carefully reading the manuscript, I found the overall description of the outcome variables highly inconsistent. I recommend that the authors take a thorough look and streamline this. For instance, they extensively discuss why a thresholded d is important. However, later, they work on $\log(t)$, which hit me by surprise. At the same time, they frequently jump back and forth between p and d .

Based on feedback from the previous round of reviews, we had dropped the predictive modeling for reporting delay proportions p in favor of focusing on event level prediction of binary indicators d and log-delays. The log-delay approach was specifically suggested by reviewers and we believe that it supplements our analysis well. Moreover, we believe that the reporting delay proportions p are still highly important for our spatial analysis: comparing the spatial variation of p and d allows us to assess the dependencies of reporting delays both within and between police districts. Nevertheless, we agree with the reviewer that the description of the variables has to be followed carefully. We have amended our manuscript to clarify their meaning and purpose better, as well as correcting a few errors where p was inadvertently used in place of d .

For a better comparison, I would suggest adding a naive baseline (majority vote) to the results tables. I assume that their outcome variable is skewed and, hence, the accuracy must be treated with caution. Here the naive baseline could facilitate a better interpretation of the results.

We agree with the reviewer that this is an appropriate and useful baseline, and have thus added it to our accuracy result tables.

Given the new positioning of the work, I can live with the feature importance. However, I am concerned with the regression results (which estimator? OLS? multicollinearity? could the latter conflate p-values? no discussion on this?). Even more important, a journal on data science like RSOS should not advocate frequentist statistics given the obvious shortcomings; instead, I would suggest a Bayesian estimator (see Gelman's BDA3 and the surrounding discussion). That being said, I would be fine with different roads that the authors could take (e.g., even removing this estimation, given that their story is on the overall predictive utility of different variable groups and, against this background, the coefficient estimates are not necessary; if they want the coefs to be included, I would advocate a plot).

We thank the reviewer for these comments. To clarify the purpose of the regression model coefficients in our paper, we see these as complementary to the variable importance analysis, since they provide the direction of effect for each variable. We agree with the reviewer on further clarification of our linear regression models, and have adapted the manuscript to clarify that we use OLS estimation and do not observe multicollinearity between the continuous predictors. Lastly, the preference between frequentist and Bayesian statistics is a matter of debate in the academic community, and thus we opted for a standard procedure well established in data analysis.

Overall, the discussion could benefit from a more truthful statement on the results: the prediction performance changes only barely across the models. Again, this is not "bad" but a finding (a relevant one) of its own. Maybe by reflecting upon the impact (e.g. how many cases would be correctly predicted additionally when changing from model NS-x to S-y) could put the results more into perspective.

We agree with the reviewer and have edited the respective summary sentence on predictive power in our Conclusion, highlighting that while improvements in predictive performance are relatively small, the more important takeaway is that our models successfully eliminate residual autocorrelation.

Reviewer: 3

The manuscript makes a clear contribution to the literature on predicting reporting delays, especially delays in rape reporting. Given its framing, which emphasizes the literature's previous inattention to spatial and temporal factors, it seems more could be said about ways in which human activities structure time and space to generate the conditions under which the forecasting model picks up these relationships. If the authors wish to put in more of this framing material, both in the introduction and the discussion, they might usefully examine the sociological literature on neighborhood effects. Likewise, the criminological literature on routine activities theory and situational crime prevention could offer some additional interpretations for both the spatial and temporal results noted.

We appreciate the reviewer's suggestion and have looked into the criminological literature on neighborhood effects, routine activities theory, and situational crime prevention respectively. We

wish to clarify that these literatures primarily relate to offender behavior and thus suggest factors that are predictive of the prevalence of rape crime (e.g., the combination of offender, target, and lack of guardianship for routine activity theory, or opportunity structure for situational crime prevention). In contrast, our paper focuses on rape reporting delays (i.e., the behavior of a survivor of rape crime), which are not well modelled by either routine activity theory or situational crime prevention. While we agree that there is a robust economic literature on neighborhood effects, and that individual reporting behaviors may well be dependent on neighborhoods, we are focused on predictive modeling rather than causal identification, and thus attempt to identify specific observable variables at neighborhood level that are predictive.